# Creating pairs of exceptional points for arbitrary polarization control: asymmetric vectorial wavefront modulation

Zijin Yang [1,2,12], Po-Sheng Huang[3,12], Yu-Tsung Lin[3], Haoye Qin[1], Jesús Zúñiga-Pérez [4,5], Yuzhi Shi[6], Zhanshan Wang[6], Xinbin Cheng [6], Man-Chung Tang [1], Sanyang Han [1], Boubacar Kanté [7], Bo Li[1,8], Pin Chieh Wu [3,9,10] ✉, Patrice Genevet [4,11] ✉ & Qinghua Song [1,8] ✉

Exceptional points (EPs) can achieve intriguing asymmetric control in non-Hermitian systems due to the degeneracy of eigenstates. Here, we present a general method that extends this specific asymmetric response of EP photonic systems to address any arbitrary fully-polarized light. By rotating the meta-structures at EP, Pancharatnam-Berry (PB) phase can be exclusively encoded on one of the circular polarization-conversion channels. To address any arbitrary wavefront, we superpose the optical signals originating from two orthogonally polarized -yet degenerate- EP eigenmodes. The construction of such orthogonal EP eigenstates pairs is achieved by applying mirror-symmetry to the nanostructure geometry flipping thereby the EP eigenmode handedness from left to right circular polarization. Non-Hermitian reflective PB meta-surfaces designed using such EP superposition enable arbitrary, yet unidirectional, vectorial wavefront shaping devices. Our results open new avenues for topological wave control and illustrate the capabilities of topological photonics to distinctively operate on arbitrary polarization-state with enhanced performances.

Hermiticity[1–4], as a physical property of conservative systems, plays an important role in quantum mechanics. Energy conservation endows Hermitian operators with real eigenvalues and orthogonal eigenstates[5,6]. However, when a system with energy gain and loss is concerned[7], the dynamics becomes non-Hermitian[8–11], characterized by the continuation of eigenvalues from the real to the complex frequency plane. Interestingly, the eigenstates orthogonality of the original Hermitian operator is broken, leading to the potential coalescence (i.e., degeneracy) of eigenstates as well as of eigenvalues at specific points, known as EPs[12–20]. Such degeneracy of eigenstates has aroused extensive interest in the asymmetric control of the eigenstates by researchers in various fields, such as mechanics[21], acoustics[22], and electromagnetism[23]. Among them, optics is considered to be an ideal platform for observing the physical phenomena corresponding to EPs.

[1]Tsinghua Shenzhen International Graduate School, Tsinghua University, Shenzhen 518055, China. [2]School of Materials Science and Engineering, Tsinghua University, Beijing 100084, China. [3]Department of Photonics, National Cheng Kung University, Tainan 70101, Taiwan. [4]Université Cote d'Azur, CNRS, CRHEA, Rue Bernard Gregory, Sophia Antipolis, 06560 Valbonne, France. [5]Majulab, International Research Laboratory IRL 3654, CNRS, Université Côte d'Azur, Sorbonne Université, National University of Singapore, Nanyang Technological University, Singapore, Singapore. [6]Institute of Precision Optical Engineering, School of Physics Science and Engineering, Tongji University, Shanghai 200092, China. [7]Department of Electrical Engineering and Computer Sciences, University of California, Berkeley, CA 94720, USA. [8]Suzhou Laboratory, Suzhou 215123, China. [9]Center for Quantum Frontiers of Research & Technology (QFort), National Cheng Kung University, Tainan 70101, Taiwan. [10]Meta-nanoPhotonics Center, National Cheng Kung University, Tainan 70101, Taiwan. [11]Physics Department, Colorado School of Mines, 1523 Illinois St., Golden, CO 80401, USA. [12]These authors contributed equally: Zijin Yang, Po-Sheng Huang. ✉e-mail: pcwu@gs.ncku.edu.tw; patrice.genevet@mines.edu; song.qinghua@sz.tsinghua.edu.cn

Recent results led to chiral absorbers that can absorb perfectly only one particular state of the input wave[24], chiral mode switching by encircling an EP[25] or asymmetric transmission of circular polarized light[26]. This asymmetry is thus endowed by the handedness of the EP (i.e., the system cannot coincide with its mirror image), which distinguishes between two opposite circularly-polarized beams. However, these efforts have always been limited to the exploitation of a single EP, restricting the response of the system to one single-polarization eigenstate defined by the handedness of the EP. In other words, using EPs and their topological properties[27] to engineer arbitrary states remains elusive.

Metasurfaces, arrays of rationally designed subwavelength micro/nano-structures that enable the flexible control of light properties[28-38], provide a great platform for such asymmetric behaviors at EPs, including asymmetric reflection and asymmetric transmission[39,40]. Specifically, a non-Hermitian metasurface, composed of sub-wavelength anisotropic structure arrays, can be designed to reach a degenerate eigenstate response at an EP, resulting in reflection singularities[41]. This peculiar behavior is extremely interesting when it occurs on the polarization states, i.e., considering an EP of the Jones matrix, because the response of the interface totally vanishes for one of the two circular polarization (CP) states, leaving the other crossed CP channel unaffected with high reflection. In essence, these reflection singularities manifest as the zero values of the off-diagonal terms in the reflection Jones matrices. Also, we note that similar behavior can be observed for transmission zeros[42]. Meanwhile, by rotating such singular meta-structure, a PB phase[43] is introduced on the CP conversion-channel unaffected by the considered EP, while the response of the system for the EP conversion-channel is by construction zero. This way, it is possible to utilize the PB phase to encode phase profile on both crossed CP independently, rotating each of the EP structure to address the phase value of the considered channel only. However, because of the intrinsic asymmetric behavior at the EP, previous systems operate only on a specific polarization state.

In this paper, we present a general method to span the effect of systems operating at EPs to any arbitrary polarization states, bypassing the intrinsic limitation of individual EP, which operates on just one defined polarization state. To illustrate our approach and confirm our theoretical treatment, we propose a plasmonic non-Hermitian metasurface[44-46] that sustains a pair of EPs in a controlled manner. Previously, in the process of reconstructing fully polarized channels[47,48] utilizing PB structures, cross-polarized twin images were always projected at the opposite diffraction angles. Here we address this issue by endowing a pair of EPs on the two channels over which the PB phase operates, producing vectorial holographic images with arbitrary spatial polarization and free of undesired twin images. To construct a pair of EPs with opposite handedness we build upon symmetry considerations, in particular mirror symmetry. Based on the asymmetric and independent reflections of the left circular-polarized (LCP) and right circular-polarized (RCP) light at each EP, an orthonormal basis of states is constructed in the polarization space, paving the way to fully control any polarization states and lifting thereby the strong practical restriction imposed by the exclusive CP response of a single EP. Moreover, by decoupling the intensity and polarization information[49,50], we utilize the plasmonic non-Hermitian metasurface to independently encode the azimuth ($\psi$) and ellipticity ($\chi$) angles of the polarization in a uniformly distributed intensity profile. The corresponding far-field polarization encoded images that have been experimentally measured do not come with the usually undesired twin images, thus solving the problem of multiple diffraction orders and background signal.

## Results
### Design method
The design principle of the topological vectorial metasurface is shown in Fig. 1, which consists of metal-dielectric-metal layers (Fig. 1a). At the top, "⌞"-shaped meta-structure (denoted as $S$) arrays are used to construct the metallic device layer. The system is excited at normal incidence using a horizontally polarized light (LP-H) impinging on the device in the -$z$ direction. Among the different structural parameters of the meta-structures (Fig. S1), $L_1$ and $L_3$ are the two main parameters varied to achieve the desired EP response. The simulated current response of the "⌞"-shaped meta-structure with $\mathbf{E}_x$ incidence is shown in Fig. 1b. The simple physical idea underlying the full-polarization control enabled by pairs of EPs consists in considering a mirror-symmetric meta-structure of the original one with respect to the $yz$ plane, i.e., an "⌟"-shape (denoted as $S^m$). In the following the superscript $m$ represents the mirror-symmetric meta-structures $S^m$ and no superscript represents the original meta-structures $S$. For each of the mirror-symmetric meta-structure $S^m$, the current distribution of the vertical arm switches the flow direction with respect to those in $S$ as shown in Fig. 1c, resulting in the flipping of the degenerate eigenstate from $\mathbf{E}_x - i\mathbf{E}_y$ to $\mathbf{E}_x + i\mathbf{E}_y$, i.e., from RCP to LCP (denoted as $|R\rangle$ and $|L\rangle$, respectively). To investigate the reflection matrix coefficients of the mirror-symmetric structure, mirror symmetric operator $\Pi$ is applied to the reflection matrix $\hat{M}$ of given structure (Supplementary Note 1). Eq. S10 shows that under the condition of CP basis, the reflection matrix coefficients of $\hat{M}$ and $\hat{M}^m$ have a strict relationship of chiral inversion, which means that the polarization conversion capability of $S$ structures on RCP light is exactly equal to that of $S^m$ for LCP light. Therefore, the existence of symmetric structures relieves the limitation of the single channel imposed by the degenerate response of a single EP, and realizes the symmetric conversion of the two CPs. Most notably, RCP and LCP can be represented by vectors $\begin{pmatrix} 0 \\ 1 \end{pmatrix}$ and $\begin{pmatrix} 1 \\ 0 \end{pmatrix}$ in the CP basis, respectively, which means that a set of orthogonal eigenvectors forming two bases in the polarization space are obtained. Full coverage of the two-dimensional polarization space can be realized by properly regulating the PB phase associated to each of the two kinds of meta-structures on each of their unaffected associated polarization-conversion channels. It is worth noting that using pair of dual EPs is not only applicable to the above photonic system, but can also be proven useful for the design of many other physical systems involving EPs, whether they display inherent chirality or not. (see more details in Supplementary Note 2).

Fig. 1d defines the parameter space of the meta-structure with mirror-symmetry. By setting $L_1$ in meta-structure $S$ (⌞) to be positive and in $S^m$ (⌟) to be negative, a summary diagram of different structures with $L_1$ and $L_3$ as $x$- and $y$-axis can be obtained. To find the EPs of the mirror-symmetric structures, CP conversion coefficients were simulated by sweeping the lengths of $L_1$ and $L_3$ and the results are shown in Figs. 1e, 1f and S4. The parameter space is set to $L_1 \in [-100, 100$ nm$]$ and $L_3 \in [50, 250$ nm$]$. Particularly, a pair of singularities induced by the EPs are obtained at $(L_1, L_3) = (\pm 52, 119$ nm$)$, and described with solid white ($M_{LR} = 0$, Fig. 1e) and black ($M_{RL}^m = 0$, Fig. 1f) stars. Two zero-reflectivity singularities constituting phase vortices with opposite topological charge are observed. The key feature underlying our strategy is that for a given structure (let's say, $S$ and its associated EP), the other polarization conversion channel (which is the coalesced eigenstate of the complementary EP, corresponding to the $S^m$ structure) does not show a singularity at that point of parameter space, as represented by the hollow stars in Fig. 1e, f. Furthermore, in the regions of parameter space corresponding to the complementary EP parameter regions of respective mirror symmetric structures (identified by the two hollow stars), the amplitude (denoted as the length of the arrows) and phase (denoted as the rotation angle of the arrows) of the CP conversion coefficients remain relatively constant and non-zero.

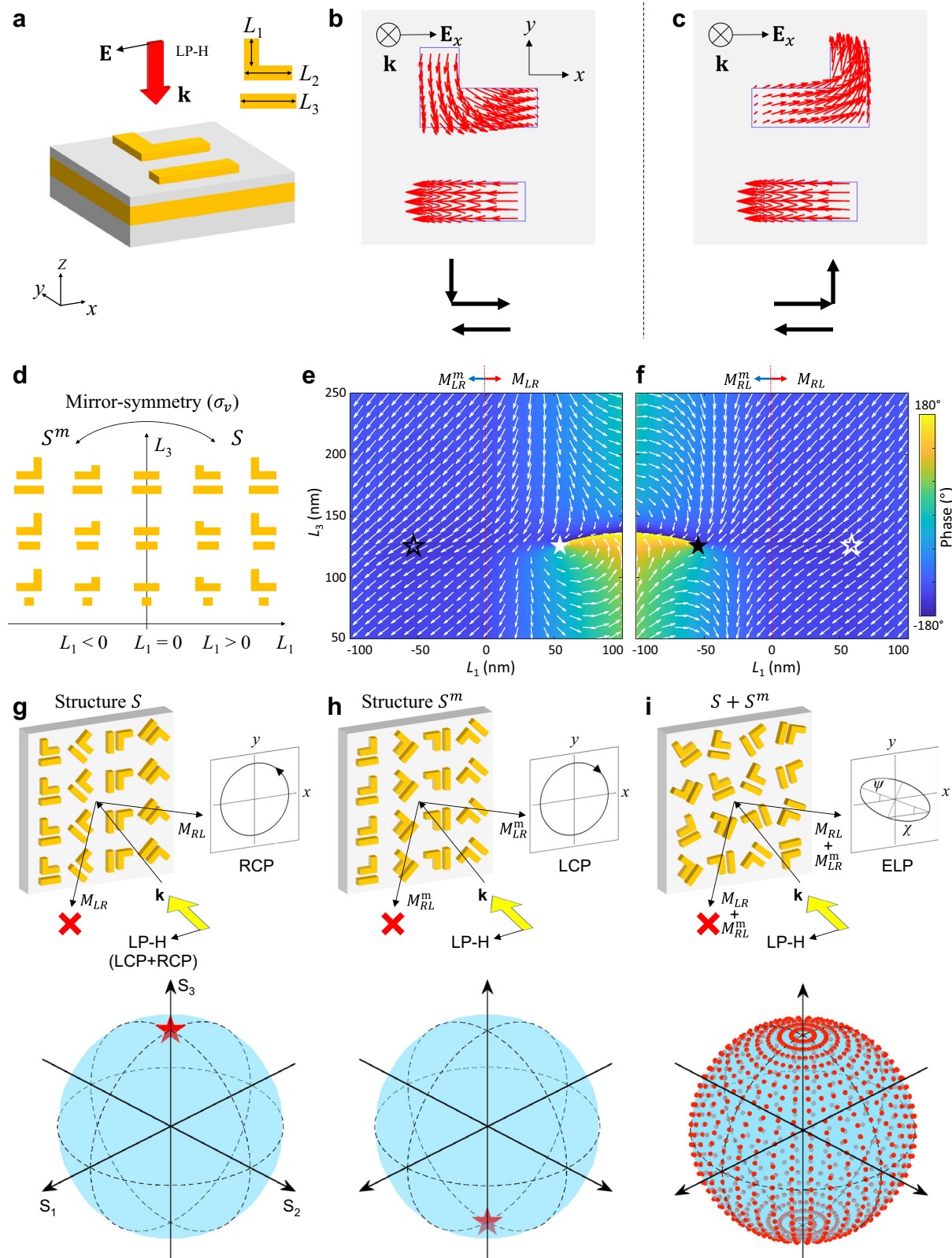

PB phase, which has been widely applied in the last years for phase modulation of CP light with metasurfaces, has opposite effects on two orthogonal circularly-polarized light beams[51]. Now that we have introduced EP pairs realized by mirror-symmetric structures, it becomes possible to engineer the phase of each CP channel independently one from the other, that is, decoupling the two CPs, while still using PB phase to reconstruct any reflective polarization channel.

According to the properties of PB phase, when the meta-structure is rotated by a certain angle $\varphi$, the incident RCP (or LCP) will be converted to the other one and will be imposed a geometric phase, i.e., $|R\rangle \rightarrow e^{-i2\varphi}|L\rangle$ and $|L\rangle \rightarrow e^{i2\varphi}|R\rangle$. When the meta-structures $S$ and $S^m$ rotate $-\varphi$ and $+\varphi$ respectively, the incident LCP (for $S$) and RCP (for $S^m$) will be converted to the corresponding cross-polarization, simultaneously adding a PB phase of $-2\varphi$. Conversely, when RCP is incident

**Fig. 1 | Design principle of topological vectorial metasurfaces with EPs pair.**
**a** Perspective view of the structural unit of topological vectorial metasurface. The incidence of horizontally linear polarized (LP-H) light is perpendicular to the metasurface along the -z direction. The inset figure on the upper right depicts the top view of the structure layer. The lengths of the three rods in the meta-structure are denoted by $L_1$, $L_2$ and $L_3$, respectively. **b, c** Simulated surface current distributions of (**b**) meta-structures $S$ and **c** $S^m$ with x-direction linear polarized incidence. **d** Top view of mirror-symmetry structures with different $L_1$ and $L_3$, where $L_1<0$ in $S^m$ and $L_1>0$ in $S$. **e, f** Complex amplitude of the simulated CP conversion coefficients in the parameter space covered by $L_1\in[-100, 100\text{ nm}]$ and $L_3\in[50, 250\text{ nm}]$, where the length and angle of the arrows represent the amplitude and phase of the complex amplitude. The solid white and black stars represent the EP pair observed for both

structures $S$ and $S^m$ respectively, while the hollow stars show the other CP conversion channel. **g, h** Schematic of the asymmetric wavefront modulation by the combination of EPs and PB phase with rotating mirror-symmetric structures. The LP-H input light can be decomposed into RCP + LCP beams. The structures $S$ in (**g**) work for LCP input light and the structures $S^m$ in (**h**) work for RCP. The polarization states of the output beams are marked on the Poincaré sphere below, where the north and south poles represent RCP and LCP, respectively. **i** Full-polarization-reconstruction through the superposition of output RCP and LCP beams at the EP pair with arbitrary phase and amplitude is realized by inputting the same LP beams. With the arrangement of two types of meta-atoms ($S$ and $S^m$) on a single meta-surface, arbitrary output polarization states can be realized.

on $S$ or LCP on $S^m$, there is no output cross-polarized beam because the conversion term ($M_{LR}$ or $M_{RL}^m$) in the corresponding Jones matrix is zero (note that this is exactly true only at the EPs). To realize asymmetric vectorial wavefront control for arbitrary polarization state reconstruction, for LCP incident light we arrange the $S$ meta-structures on a horizontal line and rotate them clockwise with the set initial orientation angle and an angle increment of $\varphi_d$ for each adjacent meta-structures, i.e., to deflect the resulting RCP output light to the right side with a deflection angle of $\theta_t$ (Fig. 1g). Similarly, for RCP incident light, by rotating the $S^m$ meta-structures counterclockwise with another initial orientation angle and setting the same angle increment $\varphi_d$, the output LCP can achieve the same deflection angle (Fig. 1h). Importantly, taken into account that any polarization of the light beams can be described by a superposition of LCP and RCP (More details in Supplementary Note 3), a single metasurface composed of two types of crossed polarized meta-atoms has been used to project holographic images at arbitrary polarization. Here, we additionally preserve the asymmetric wavefront operation characteristics of EP designs by choosing two symmetrical EP meta-atoms, each respectively operating at one of the two orthogonal polarization states (Fig. 1i). By encoding the holographic phase information on both LCP and RCP beams starting from the initial orientation angle and applying a modified Gerchberg–Saxton (GS) algorithm[48], arbitrary polarized meta-hologram can be achieved. It is worth noting that, due to the symmetric processing capability of the PB phase for circularly polarized light, the other CP beam generally appears in the $-\theta_t$ direction. However, the introduction of an EP pair totally suppresses redundant images in unnecessary directions, yielding to asymmetric meta-hologram with single order. In brief, the design method uses the PB phase to realize the vectorial holographic images while avoiding unnecessary orders thanks to the degeneracy of the EP, providing a general strategy for asymmetric control of the vectorial wavefront.

The simulation results of the EP pair are shown in Fig. 2. Two singularity points in the corresponding reflection matrices of the mirror-symmetric structures, where $M_{LR}=0$ and $M_{RL}^m=0$, are observed simultaneously at $\lambda=600$ nm (Fig. 2a). On the contrary, at those same points $M_{RL}$ and $M_{LR}^m$ remain stable without significant change. Fig. 2b, c shows the corresponding eigen polarization states on the Poincaré sphere for the meta-structures $S$ and $S^m$, respectively, with the incident wavelength ranging from 550 to 650 nm. The eigenstates associated to each of the structures degenerate as RCP (Fig. 2b) and LCP (Fig. 2c), respectively, at $\lambda=600$ nm, proving the existence of an EP pair. For in-depth understanding of the eigenstates at the wavelength of 600 nm, the azimuth and ellipticity angles of eigenstate 1 and 2 in the parameter space covered by $L_1\in[-100, 100\text{ nm}]$ and $L_3\in[50, 250\text{ nm}]$ are shown in Fig. 2d–g. In the EP pair, i.e., $L_1=\pm52$ nm and $L_3=119$ nm, the azimuth and ellipticity angles of eigenstate 1 and eigenstate 2 degenerate. Stars indicate the position in the parameter space where EPs emerge. The white and black stars represent the degeneracy of two eigenstates into RCP and LCP, respectively. All the above simulations confirm that a pair of EPs are obtained at $L_1=\pm52$ nm and $L_3=119$ nm.

## Asymmetric and arbitrarily-polarized meta-hologram

To verify the design principle, we demonstrate four meta-holographic images with different polarization states using a pair of EPs combined with PB phase. Simulation results of the CP conversion efficiency and relative PB phase of the metasurfaces are shown in Fig. 3a, b. When structures $S$ and $S^m$ are rotated, only one of the polarization channels is converted (LCP for $S$ and RCP for $S^m$), and the corresponding PB phase is introduced. The conversion efficiencies of the two symmetric structures are both about 20% in the above available channels, and the phase coverage can reach 2π. Meanwhile, as requested by the handedness of the associated EP, the conversion coefficient of the other channel is near-zero. In the experiments, the horizontally polarized light (corresponds to x-polarization, LP-H), which can be viewed as a superposition of LCP and RCP, was generated by a linear polarizer and used as the incident beam. By changing the number of rows of $S$ and $S^m$ to adjust the amplitude, and the rotation angle between the two rows to introduce the phase difference between LCP and RCP components (More details in Supplementary Note 3), we realized the holographic images with LCP (Fig. 3c), RCP (Fig. 3d), linear polarization (LP, Fig. 3e) and elliptical polarization (ELP, Fig. 3f). According to the formula:

$$\theta_t = \arcsin\left(\frac{2\varphi_d}{k_0 p}\right) \tag{1}$$

The holographic images are set to 30° by controlling the orientation angle increment $\varphi_d=45°$ (where $k_0$ is the wavenumber in the free space and $p$ is the period of the unit-cell). As expected, four different holographic images (L, R, A and H letters) with designed polarization states clearly appeared at the angle of 30° when LP-H light was incident. More importantly, no redundant images appeared at -30° (some faint and incomplete images are observed due to the fabrication imperfection, to be compared with completely equivalent images when no EP is used), proving the realization of the asymmetric control of the meta-hologram. The polarization states of the resulting holographic images are shown in Fig. 3g, where the blue and red dots on the Poincaré sphere represent the designed and measured polarizations, respectively. Their close positions indicate that the experimental results are in great agreement with theoretical expectations.

## Asymmetric vectorial meta-hologram

In the following, we independently encoded the azimuth and ellipticity angles of the polarization information with controllable amplitude distribution, known as vectorial meta-hologram. A pixel containing two rows of rotated mirror-symmetric structures is designed to achieve arbitrary combination of spatially varying amplitude and polarization information, as shown in Fig. 4a. By keeping the rotation angle of the top and bottom rows opposite to each other, so as to impose equal but opposite sign of phase gradients, RCP and LCP beams will be generated at the same angle of $\theta_t$. Fig. 4b shows the SEM image of the fabricated metasurface. The intensity of the profile is set to a constant value in the

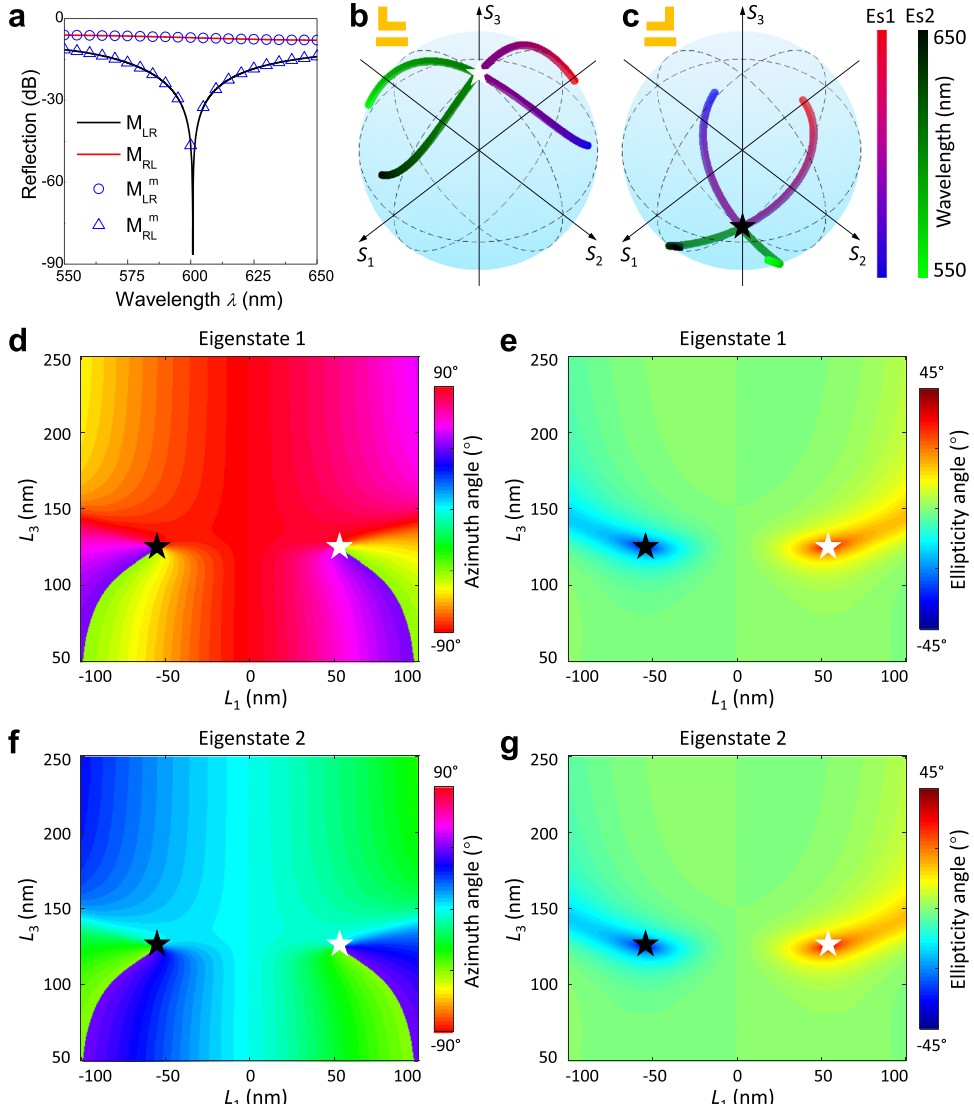

**Fig. 2 | Simulation results of the EP pair with the mirror-symmetric structures.** **a** Spectral dependence of the reflection coefficients in the CP basis. Two zero-reflectivity singularity points with $M_{LR} = M_{RL}^m = 0$ are observed at $\lambda = 600$ nm. **b**, **c** The eigenstates of (**b**) meta-structures $S$ and (**c**) $S^m$ projected on the Poincaré sphere as a function of wavelength, showing that they degenerate at the two poles (RCP and LCP) at $\lambda = 600$ nm, where Es1 and Es2 represent eigenstate 1 and

eigenstate 2. **d**–**g** Eigenstates of the reflection matrices in parameter space $(L_1, L_3)$ at a wavelength of 600 nm. **d** Azimuth and **e** ellipticity angles of the eigenstate 1. **f** Azimuth and **g** ellipticity angles of the eigenstate 2. The parameter space is set to $L_1 \in [-100, 100$ nm] and $L_3 \in [50, 250$ nm], where white and black stars represent EPs in meta-structures $S$ and $S^m$, respectively.

region of interest. Then, according to the formula: $\psi = \delta$ and $\chi = \frac{1}{2}\arcsin\frac{a_R^2 - a_L^2}{a_R^2 + a_L^2}$, by manipulating the amplitude ($a_L$, $a_R$) and phase difference ($\delta$) of the two CPs over the far-field spatial distribution, arbitrary combinations of ellipticity and azimuth angles can be achieved in the selected region (Supplementary Note 4). The modified Gerchberg-Saxton (GS) algorithm[48] of vector field is used to realize the decoupling of amplitude and far-field polarization information. It is noteworthy that because the number of rows of $S$ and $S^m$ in the meta-surface is equal, there is no additional degree of freedom to control the intensity difference between LCP and RCP. As compensation, the azimuth and ellipticity angles need to be tuned (translated and scaled) to make the total strength of LCP and RCP equal in the far field. Simulation results of the distribution of intensity, azimuth and ellipticity angles in the far-field are shown in Figs. S5a−S5c. A circular bright spot like the sun confirms the uniform distribution of intensity, while the images of "Infinity" (Fig. S5b) and "Tai Chi" (Fig. S5c) represent the azimuth angle and ellipticity angle, respectively. Measured intensity results are shown

in Fig. 4c with designed vectorial holographic image at 30° but no image at -30°. The decoding of the polarization information is based on the Stokes parameters[52], which can be obtained by introducing a quarter-wave plate and a linear polarizer in the optical setup. As expected, a circular intensity distribution (Fig. 4d) with spatial distribution of polarization in azimuth angles of "Infinity" (Fig. 4e) and ellipticity angles of "Tai Chi" (Fig. 4f) are displayed respectively. Furthermore, grayscale images were used to demonstrate the enormous potential of the Modified GS algorithm. In Fig. S11, three grayscale images are selected as intensity, azimuth and ellipticity angles respectively, and the corresponding simulation results show that the meta-holograms are well reconstructed, indicating that the algorithm can achieve arbitrary combinations of intensity, azimuth and ellipticity angles in the selected area. It should be noted that the asymmetric vectorial hologram can be designed to provide broadband response since the channels $M_{RL}$ and $M_{LR}^m$ are always larger than $M_{LR}$ and $M_{RL}^m$ in a wide frequency range as shown in Fig. 2a.

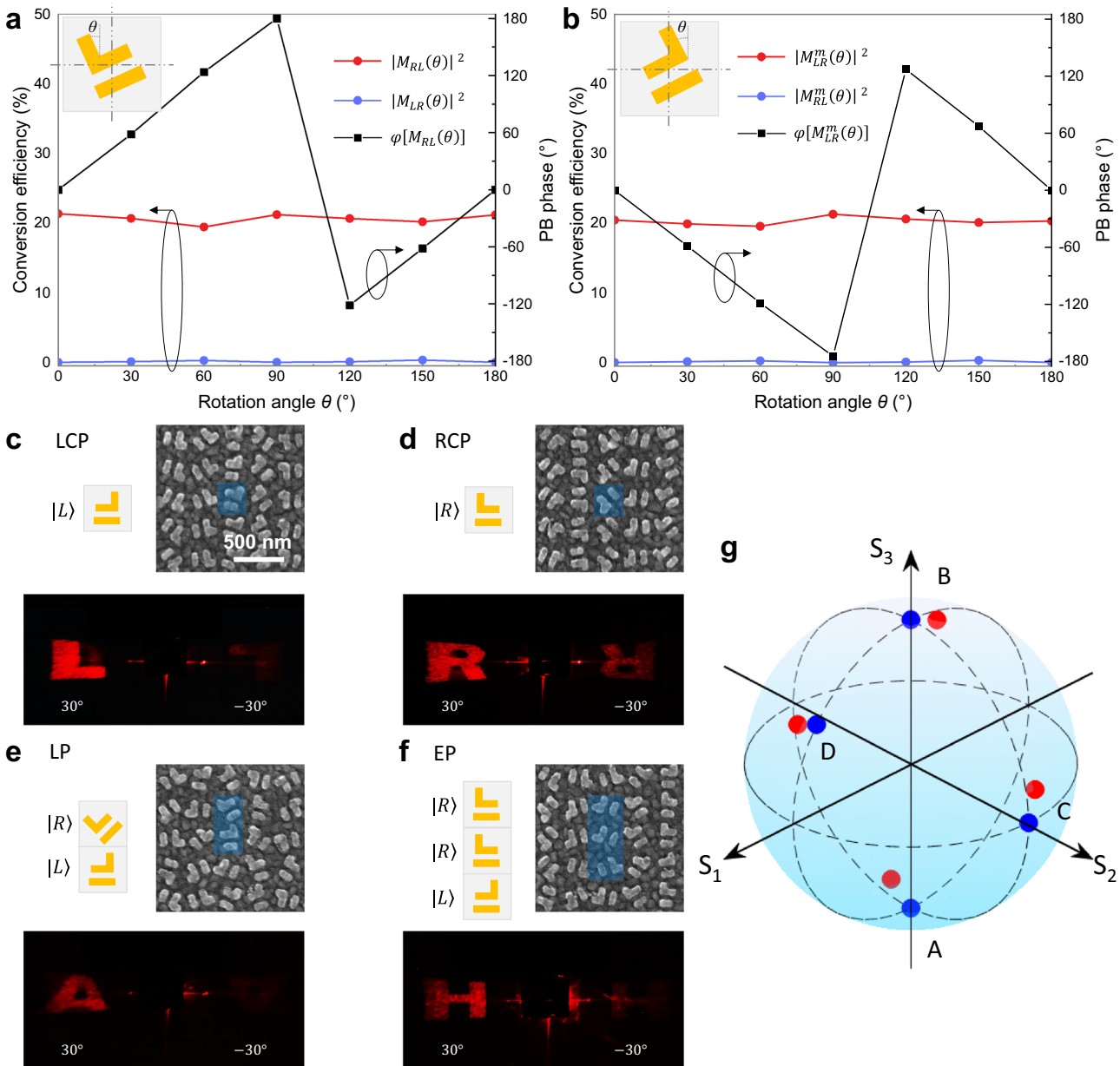

**Fig. 3 | Asymmetric meta-hologram with different polarization designs.** Simulation results of the CP conversion efficiency and relative PB phase of the metasurfaces when the meta-structures of (**a**) $S$ and (**b**) $S^m$ rotate at different angles. **c–g** Fabricated and measured results of the asymmetric meta-holograms. The polarizations of the holograms were designed as **c** LCP ($\chi = -45°$), **d** RCP ($\chi = 45°$), **e** LP-45° ($\psi = 45°$, $\chi = 0°$) and **f** ELP ($\psi = 0°$, $\chi = 18.43°$). Asymmetric holographic images can be seen with clear images of "L", "R", "A" and "H" at 30° but only faint and incomplete images appear at the opposite angle of −30°. **g** The corresponding polarization of the holographic images on Poincaré sphere. Blue dots: designed polarizations. Red dots: experimentally measured polarizations.

## Discussion

In conclusion, we have demonstrated a general approach to extend the application of EPs to any arbitrary polarization states, thereby surpassing the inherent limitation of the degenerate response that EP brings to the system. Using a plasmonic non-Hermitian metasurface as the experimental platform and thanks to the strategy of mirror-symmetric transformation, the inversion of an EP handedness is achieved. The design flexibility of metasurfaces enables combining both EPs generated from a meta-structure and its mirror-image to generate a pair of EPs with opposite phase circulation singularity. Thanks to the EP pair we implemented the asymmetric reflection of the two CP beams, extending the metasurface response to all polarization states beyond the CP states. The combination of

an EP pair with the PB phase is shown to suppress unnecessary redundant images that limits the technological use of PB-phase metasurfaces while achieving full polarization reconstruction. To demonstrate the multi-channel multiplexing capability of the non-Hermitian metasurfaces, we encoded complex polarization information into the intensity distribution. Overall, this work breaks the inherent limitation of topological metasurfaces, improving the possibility of topological wavefront-shaping for wide applications, and opens the way for optical communications, data encryption, and high-density optical data storage with topological metasurfaces. We believe that the idea of coupling a pair of EPs to realize the expansion of the asymmetric response can be applied to a wider range of EP systems.

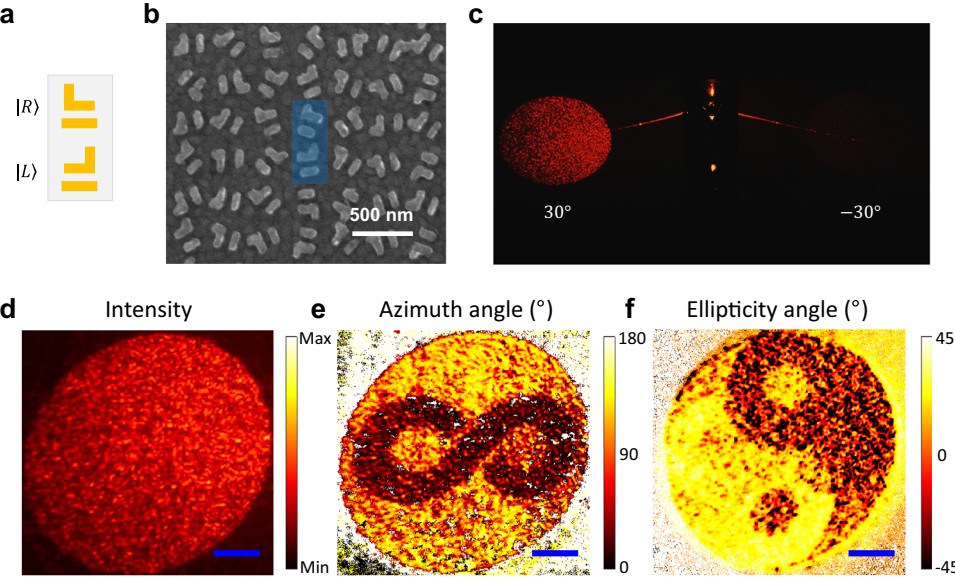

**Fig. 4 | Asymmetric vectorial meta-hologram with spatially distributed polarization profile. a** Top view of the structure design with the combination of $S$ and $S^m$. **b** SEM image of the fabricated metasurface. **c** The full-view of the intensity distribution of the measured meta-hologram. A clear image is shown at 30° but no image at −30°. **d–f** Measured vectorial meta-hologram at 30° with distribution of **d** uniform encoded intensity, **e** spatially distributed azimuth ("Infinity") and **f** ellipticity ("Tai Chi") angles of the polarization in the far-field. The blue scale bar in **d-f** represents 1 cm.

## Methods

### Device fabrication

The fabrication processes are shown in Fig. S2. The cleaned silicon wafer is diced into $1 \times 1$ cm$^2$ square substrates. Then thermal evaporation is used to grow an aluminum (Al) layer with a thickness of 150 nm on the substrate as metal ground. Subsequently, SiO$_2$ spacer layer of 40 nm is deposited on the Al layer by magnetron sputtering. Whereafter, ~120 nm thick CSAR 62 resist is spin-coated on the spacer layer and then baked at 180 °C for 90 seconds. Furthermore, electron-beam lithography (Elionix ELS-7500) exposure is performed at 50 kV, while the structural design is made by BEAMER-GenISys software. After development of CSAR 62 with ZED-N50 solution for 4.5 minutes, IPA fixing process is performed for 30 seconds. Next, 30 nm Al layer is deposited using thermal evaporation. Finally, the liftoff process is operated using Remover PG. Followed by IPA and deionized water rinsing, the metasurface is created.

### Optical setup

Schematic diagram of the optical setup is shown in Fig. S3. A laser beam at a wavelength of 600 nm passes through a linear polarizer with a horizontal transmission axis, therefore, a LP-H beam is obtained. Then the beam propagates through a lens and is weakly focus on the meta-hologram. Since the system works in reflection, projector and incident beam are found on the same side of the metasurface. In addition, there is a hole in the center of the projector to allow incident light to pass through. The detection of the polarization state of the beam is done by a polarimeter (Thorlabs PAX1000VIS).

In order to characterize the polarization information in the far-field, a quarter-wave plate and a linear polarizer are placed in front of the holographic image. By adjusting the rotation angle of the linear polarizer and controlling the addition/removal of the quarter-wave plate, the intensity values under different polarization filtering conditions can be obtained. The corresponding azimuth and ellipticity angles can be obtained after post-processing.

## Data availability

The data that support the findings of this study are available from the corresponding author upon reasonable request.

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

## Acknowledgements

Q.S. acknowledges the funding support from the National Key R&D Program of China (No. 2023YFB3811400), the National Natural Science Foundation of China (No. 12204264), and the Shenzhen Science and Technology Innovation Commission (No. WDZC20220810152404001; JCYJ20230807111706014). Q.S. and S.H. acknowledges the funding from the Cross-disciplinary Research and Innovation Fund of Tsinghua Shenzhen International Graduate School (No. JC2022001). Y.S. acknowledges the funding support from the National Natural Science Foundation of China (No. 62205246), Shanghai Pilot Program for Basic Research, Science and Technology Commission of Shanghai Municipality (22ZR1432400), and the Fundamental Research Funds for the Central Universities. B.K. acknowledges support for the U.S. Army Research Office (ARO) grant W911NF2310027. P.C.W. gratefully acknowledges the use of advanced focused ion beam system (EM025200) of National Science and Technology Council (NSTC) 112-2731-M-006-001 and electron beam lithography system belonging to the Core Facility Center of National Cheng Kung University (NCKU). P.C.W. acknowledges the support from the NSTC, Taiwan (Grant number: 111-2112-M-006-022-MY3; 111-2124-M-006-003; 112-2124-M-006-001), and in part from the Higher Education Sprout Project of the Ministry of Education (MOE) to the Headquarters of University Advancement at NCKU. P.C.W. also acknowledges the support from the MOE (Yushan Young Scholar Program), Taiwan and supported in part by Higher Education Sprout Project, Center for Quantum Frontiers of Research & Technology (QFort) at NCKU.

## Author contributions

Q.S. and P.G. conceived the ideas and designed the research; Z.Y. and Q.S. performed the numerical calculations; P.S.H., Y.T.L. and P.C.W. fabricated the devices; Z.Y. performed the measurement; Z.Y., J.Z.P., P.C.W., P.G. and Q.S. wrote the manuscript; Q.S., P.C.W. and P.G. supervised and coordinated the project; Z.Y., H.Q., J.Z.P., Y.S., Z.W., X.C., M.C.T., S.H., B.K., B.L., P.C.W., P.G. and Q.S. discussed the results and all authors approved the paper.

## Competing interests

The authors declare no competing interests.
