## [Peer Review File · Nature Communications]

REVIEWER COMMENTS

Reviewer #1 (Remarks to the Author):

This manuscript reports on the design and implementation of a plasmonic metasurface that achieves asymmetric reflection of circularly-polarized light beams. The main idea behind the design is the use of a pair of loss-induced exceptional points (EPs) in the Jones matrix.

The technical basis of the work, both experimental and theoretical, appears to be sound. However, I am not convinced that the results are significant enough to warrant publication in Nature Communications. The general idea of using EPs to design metasurfaces, including achieving polarization control, has been previously studied (as acknowledged in the introduction). The ideas in the present paper, while novel, seem to represent a fairly incremental technical advance over what has come before.

Throughout the paper, the statements of novelty are jargon-heavy and quite narrow: "a general method to span the use of EP-induced singularities beyond the single degenerate eigenstate"; "combining PB-phase wave control of a pair of cross-polarized EPs eigenstates," "a general approach to extend the application of EPs to any arbitrary polarization state, surpassing thereby the inherent limitation of unique CP response of a system operating at an EP". One of the specific new behaviors identified by the authors is that whereas previous metasurfaces "have been used to reconstruct fully polarized channels, but cross-polarized twin images were always projected at the opposite diffraction angle," their new metasurface can suppress such twin images.

Such selling-points are scientifically valid, but I do not think they are of sufficiently broad interest. The paper does not present these issues to be particularly important either technologically (e.g., key bottlenecks for practical applications of metasurfaces; are the above twin images a critical limitation for anything?), or theoretical (e.g., capabilities that are fundamentally impossible to achieve via other design approaches). These are aspects required for high-impact general-purpose journals like Nature Communications.

Another issue with the presentation is that the connection between the physics of EPs and the metasurface-design strategy is quite hard to tease out. The problem starts with the second sentence of the abstract: "Because of their intrinsic chiral asymmetric responses, a given EP operates on only one and well-defined polarization channel, a specific property that enables intriguing asymmetric light control." Actually, this is NOT a general property of EPs, only a specific class of EPs in a certain narrow context (metasurfaces). EPs are mathematical constructs that don't have any intrinsic relation to physical concepts of "chirality" or "polarization"; even in other optics settings, EPs don't need to be related to polarization.

The first paragraph of the paper has a similar problem; it talks about both the general properties of EPs and the use of EPs in optics, but lacks a good logical flow from the general (mathematical) to the specific (light control). The paragraph also refers to concepts like EP handedness and EP order without proper introduction.

Later, in the "design methods" section, the role of the EPs does not stand out. The use of the EPs is embedded in a mass of details including the description of structural parameters, definition of the R matrix, and mirror symmetry analysis, which make the key ideas hard to stand out. I can basically follow what is going on, after multiple readings, but the presentation really ought to be improved.

All that said, I think the method and results of the paper are valid, and would be of interest to subject matter experts. I think the paper could be published, with little modification, in a more specialized journal.

Reviewer #2 (Remarks to the Author):

The manuscript reports the experimental demonstration of arbitrary polarization generation by using a pair of exceptional points (EPs). EPs are nanostructures that has non-zero conversion efficiency from LCP to RCP but zero from the opposite conversion (structure S). For a planar chiral structure, a mirror-symmetry one works opposite (structure S^m). Therefore, RCP component is purely converted from S and LCP component is purely converted from S^m with linearly polarized incidence. Arbitrary polarization and holographic images can be generated by multiplexing structures S and S^m and PB phase.

Holographic images with different polarization and vectorial wave front shaping are demonstrated with EPs. The manuscript demonstrates novel results. The theory in Supplementary Information well supports their design and experimental results. Overall, the manuscript can be accepted for publication after a minor revision considering the following comments.

1. There are too many symbols, and they might make readers confused. For example:
 - a. Azimuth and ellipticity angles: They repeat in the whole manuscript. The authors should note the symbol the first time mentioned them (lines 97 and 98).
 - b. Symbol for azimuth angle is ψ (psi) in line 248 but is φ (phi) in line 447.

c. There are too many φ s (e.g., orientation angle φ , initial orientation angle φ_{\pm} , angle increment φ_d) and might not be necessary. I suggest the authors not use symbols for those only mention once or twice.

d. Every time I read $|-\rangle$ and $|+\rangle$, I cannot link the results with RCP and LCP by intuition. I suggest the authors replace them with $|R\rangle$ and $|L\rangle$.

e. The matrix R can be replaced by M (refers to “m”etasurface or “m”atrix) so that it would not be confused with RCP.

3. Dose the initial orientation angle for S and S^m affect the polarization state in the far field (e.g., results in Figs 1I, 3E and 3F) if multiplexing a pair of EPs or more?

4. Figure captions of Figs 3E and 3F shows the corresponding azimuth and ellipticity angles. However, there is no mention in the main text. Could the author explain more about how to design specific polarization of output holographic images?

5. There are no explanation of the abbreviation “Es1” and “Es2” shown in Fig 2c.

6. The authors demonstrate 2 eigenstates degenerate at RCP and LCP. Is there any limitation to degenerate 2 eigenstates at arbitrary elliptical polarization?

7. In lines 227-229, the authors address that “The holographic images are set to 30° by controlling the orientation angle increment $\varphi_d = 45^\circ$.” Could the authors explain more about the relation between orientation angle increment and the angle the holographic images reconstructed.

8. In lines 253-255, the authors address that ‘the images of “Infinity” (Fig. S4B) and “Tai Chi” (Fig. S4C) represent the azimuth angle and ellipticity angle, respectively.’ Does the “intensity distribution” of the reconstructed images represent the value of spatial distribution of azimuth/ellipticity angle? What are the values of azimuth/ellipticity angle we can be observed from the image?

9. Both images of “Infinity” and “Tai Chi” are binary (either 1 or 0). Could the authors achieve arbitrary combinations (e.g., greyscale images) of ellipticity and azimuth angles in the selected region, as mentioned in line 250-251?

10. There are some famous literatures about demonstration of independent phase profile for RCP and LCP (or arbitrary orthogonal elliptical polarizations) incident light [Nat. Nanotech. 10, 937-943 (2015) &

Phys. Rev. Lett. 118, 113901 (2017)]. They utilize both propagation phase and PB phase of nanostructures in the design. Therefore, there is no necessary to have orthogonal EP eigenstates pairs in the design. Could the authors compare the difference and address their advantages?

Reply report to reviewer #1

Comment 1.1: *This manuscript reports on the design and implementation of a plasmonic metasurface that achieves asymmetric reflection of circularly-polarized light beams. The main idea behind the design is the use of a pair of loss-induced exceptional points (EPs) in the Jones matrix. The technical basis of the work, both experimental and theoretical, appears to be sound.*

Response 1.1: We would like to thank the reviewer for the careful reading and its positive evaluation of the technical basis of our work.

Comment 1.2: *However, I am not convinced that the results are significant enough to warrant publication in Nature Communications. The general idea of using EPs to design metasurfaces, including achieving polarization control, has been previously studied (as acknowledged in the introduction). The ideas in the present paper, while novel, seem to represent a fairly incremental technical advance over what has come before.*

Response 1.2: The authors thank the reviewer for the affirmation of the novelty of our ideas. Indeed, as mentioned by the referee and presented in the introduction of the current work, there are some pioneer works using EPs to achieve polarization control, such as *PRL* 113, 093901 (2014) and *Science* 373, 1133-1137 (2021). However, because these works employ a single EP, and because each EP is associated to a given circular polarization state (defined by the handedness of the chosen EP), these works were essentially limited to working with one unique polarization state, which needed to be furthermore circularly polarized. Indeed, for many years it seemed that there was no way out of this dilemma: EPs can only be exploited with CP states of defined chirality. In this paper we show that one can get rid of this seemingly intrinsic limitation of EPs through the following idea: thanks to the large design space of metasurfaces, one can fabricate a metasurface displaying simultaneously two spectrally-degenerate EPs of opposite handedness. Then, because any polarization state can be decomposed into the orthonormal polarization basis defined by the two opposite CP states, one can finally engineer the response of any arbitrary polarization state thanks to the two spectrally-degenerate EPs.

In addition, given that each EP affects the phase of polarization states of only one handedness and leaves almost unaffected the phase of states displaying opposite handedness, two independent exceptional topological phases for the two CP light can be achieved, enabling the design of vectorial holographic images, having promising applications in the vectorial beam generation and providing a new paradigm for a variety of polarization-encrypted nanophotonic applications with drastically enhanced capacity and security.

Nevertheless, the referee's comment shows that we need to be more pedagogical, highlight the originality of our idea and illustrate clearly its potentiality beyond the particular examples of polarization control demonstrated in the original manuscript. Thus, to prove the universality of the EP pairs idea, we have in this revision extended

the idea to other systems capable of supporting simultaneously EPs of opposite handedness thanks to the strategy introduced in our paper. We have added a new long section (**Supplementary Note 2**) in the **Supplementary Information**. It is noteworthy that the universality of our approach allows us to expand its utility to fields beyond photonics, including acoustic and quantum systems, as discussed below.

To illustrate the usefulness of our strategy and its much wider applicability, compared to other strategies, we have analyzed its advantages with respect to 3D chiral structures. In their manuscript, Tong et al.¹ have realized acoustic circular dichroism (CD) effect using three-dimensional (3D) chiral metamaterials that support circularly polarized transverse sound. They found that in systems with C_4 symmetry, the acoustic CD effect is almost negligible. However, after selectively increasing the loss of part of the unit cell, the rotational symmetry of the system decreases from C_4 to C_2 , and the CD effect is strongly enhanced. Based on their theory, they prepared a metamaterial with a three-dimensional chiral structure and achieved selective absorption of the RCP sound. However, due to the inherent chirality of the system, this metamaterial does not have the ability to process LCP sound, resulting in inability to expand to arbitrary polarization absorption. What's more, Wang et al.² demonstrated a non-Hermitian acoustic metasurface mirror with extremely asymmetric reflection at EPs. The mirror is designed to have high-efficiency retroreflection when the wave comes from one side, and nearly perfect absorption when the wave comes from the other side (i.e., unidirectional reflectionless propagation). And similar studies have been conducted in optical and microwave systems^{3,4}. However, only using a single EP makes these designs not capable of multi-directional processing. And the properties of the unidirectional reflectionless propagation are limited to one side of the device.

Our approach can overcome these and other limitations, as explained now in **Supplementary Note 2**.

“Supplementary Note 2: Universality of the concept of EP pairs

It is worth noting that the proposed idea of generating EP pairs and use them to control the polarization of light beams can be applied to a wide range of EP systems, whether chiral or non-chiral. To prove the universality of the idea of EP pairs, we propose its use in several different EP systems as shown in **Fig. S7**:

Supplementary Figure 7. Application of the concept of EP pair in other EP systems.

The ideas adopted in this manuscript can be extended to other EP systems, whether chiral¹⁻³ or non-chiral⁴⁻⁶. The diagram shows a variety of EP systems that could potentially enable new applications, provided those systems operate with a pair of EPs with opposite helicity.

Two of these examples are analyzed in detail:

(1) In the system designed by Tong et al.¹, the prepared three-dimensional chiral acoustic metamaterials (**Fig. S8A**) can achieve selective absorption of RCP sound. To construct a pair of EPs, we performed a mirror symmetry operation on the structure in **Fig. S8A** to obtain the structure shown in **Fig. S8B**. It can be found that they both have 3D chirality. To study the corresponding working modes of the two mirror-symmetry structures, their sound processing capabilities were analyzed individually. To better study the response of the 3D structures, we divide them into 2D slides and analyze the propagation across the overall system:

$$\begin{cases} M_{all}^A = M_1^A \cdot M_2^A \cdot \dots \cdot M_n^A \\ M_{all}^B = M_1^B \cdot M_2^B \cdot \dots \cdot M_n^B \end{cases} \quad (1-1)$$

where M_{all}^A and M_{all}^B represent the control matrices of the entire structure A and B, respectively, and M_n^A and M_n^B represent control matrix of the n th layer. By considering infinitesimal slices (i.e., by considering sufficiently large number n of 2D sections), M_n^A and M_n^B can be simplified into planar control matrices, just like Jones matrices.

$$M_n^A = \begin{bmatrix} M_{xx}^{An} & M_{xy}^{An} \\ M_{yx}^{An} & M_{yy}^{An} \end{bmatrix} \quad (1-2)$$

$$M_n^B = \begin{bmatrix} M_{xx}^{Bn} & M_{xy}^{Bn} \\ M_{yx}^{Bn} & M_{yy}^{Bn} \end{bmatrix} \quad (1-3)$$

Based on our derivation in **Supplementary Note 1**, it can be obtained:

$$\begin{cases} M_{xx}^{An} = M_{xx}^{Bn} \\ M_{xy}^{An} = -M_{xy}^{Bn} \\ M_{yx}^{An} = -M_{yx}^{Bn} \\ M_{yy}^{An} = M_{yy}^{Bn} \end{cases} \quad (1-4)$$

On the circular polarization basis:

$$\begin{pmatrix} M_{LL}^{Bn} & M_{LR}^{Bn} \\ M_{RL}^{Bn} & M_{RR}^{Bn} \end{pmatrix} = \begin{pmatrix} M_{RR}^{An} & M_{RL}^{An} \\ M_{LR}^{An} & M_{LL}^{An} \end{pmatrix} \quad (1-5)$$

The results show that the response of structure A to the RCP will be completely equivalent to the response of structure B to the LCP.

To better understand the coupling between the LCP and RCP modes, we construct the Hamiltonian describing the above system. For C_4 systems:

$$H_{C_4} = \begin{bmatrix} \omega_0 - i\gamma & (v_R + iv_I)k \\ (v_R + iv_I)k & \omega_0 - i\gamma \end{bmatrix} \quad (1-6)$$

where ω_0 is the resonance frequency, k is the wave number, v_R and v_I are the real and the imaginary parts of the complex group velocities, and γ is the loss.

For C_2 systems, the loss is selectively added, resulting in a symmetry-breaking, which can be described by the perturbation term $\frac{1}{2}\delta$ in the Hamiltonian. In addition, LCP and RCP modes now have different losses γ_1 and γ_2 . Therefore, the extended Hamiltonian is:

$$H_{C_2} = \begin{bmatrix} \omega_0 - i\gamma_1 + \frac{1}{2}\delta & (v_R + iv_I)k \\ (v_R + iv_I)k & \omega_0 - i\gamma_2 - \frac{1}{2}\delta \end{bmatrix} \quad (1-7)$$

It is worth noting that the introduction of the perturbation term and the unequal losses make the elements of the main diagonal, which represent the two modes, no longer equivalent. When the mirror symmetry operator acts on M_{all}^A , these two elements of the Hamiltonian will be interchanged,

$$H_{C_2}^{mirror} = \begin{bmatrix} \omega_0 - i\gamma_2 - \frac{1}{2}\delta & (v_R + iv_I)k \\ (v_R + iv_I)k & \omega_0 - i\gamma_1 + \frac{1}{2}\delta \end{bmatrix} \quad (1-8)$$

which indicates that there will be a mode transition in the symmetric structure, that is, the two symmetric structures will work in two different modes (RCP and LCP), as shown in **Fig. S8**. Briefly speaking, through a simple symmetric operation, we have implemented a pair of EPs using two 3D chiral structures. These two structures can selectively absorb RCP and LCP sound, respectively. Due to the ability to process orthogonal polarization states, the absorption of fully polarized sound can be achieved by properly combining structures A and B, greatly expanding the application range of the system.

Supplementary Figure 8. Implementation of EP pairs in three-dimensional chiral structures. (A) The schematic diagram of the unit cell of the 3D acoustic metamaterial. Each unit consists of six chiral resonators with a structure similar to that of a fan. Orange and green blades have different connection points, so the two are not equivalent. The depth of color indicates the inclined state of the fan blade. To reduce the symmetry from C_4 to C_2 , the resonators on both sides perpendicular to the x -axis are individually increased in loss. The structure operates in RCP mode, and the original structure comes from Tong et al¹. (B) Schematic diagram of the unit cell of A's symmetrical structure. This structure operates in LCP mode.

(2) Another application of our strategy consists in expanding unidirectional reflectionless propagation to non-chiral systems. A ring-square structure was constructed, as shown in Fig. S9A. The corresponding scattering properties of the system are given by the transfer matrix^{7,8}:

$$T_{all} = T_{ring} \cdot T_{propagation} \cdot T_{square} = \begin{bmatrix} T_{11} & T_{12} \\ T_{21} & T_{22} \end{bmatrix} \quad (1-9)$$

And the transmission and reflection coefficients can be defined by Eq. 1-10, as⁵

$$t = t_{12} = t_{21} = \frac{1}{T_{22}}, \quad r_{22} = \frac{-T_{21}}{T_{22}}, \quad r_{11} = \frac{T_{12}}{T_{22}} \quad (1-10)$$

It leads to the scattering matrix S describing the optical performance of the system given as:

$$S = \begin{bmatrix} t_{12} & r_{11} \\ r_{22} & t_{21} \end{bmatrix} \quad (1-11)$$

where the subscript 1 and 2 represent $+z$ and $-z$ directions of the metamaterials respectively; that is, t_{12} represents the incident beam transmitted from the top to the bottom of the material, and r_{11} represents the incident beam from the top reflected towards the space above the structure.

In the above dual-ring structure, $r_{11} = 0$ can be achieved by properly adjusting the structural parameters (note that $t_{12} = t_{21} \neq 0$, $r_{22} \neq 0$ at this time), thus reaching EP (Fig. S9B). When the mirror symmetry operation is applied to this structure, the off-

diagonal terms are exchanged as analyzed in **Supplementary Note 1**. Under these circumstances, $r_{22} = 0$, indicating that the reflection in the opposite direction is suppressed, and a pair of EPs is formed (**Fig. S9C**).

Fig. S10 illustrates how to achieve extended unidirectional reflectionless propagation by combining a pair of EPs. As shown in **Fig. S10A** and **S10B**, mirror-ring symmetry operations do not disrupt the reciprocity of the transmission, so distortion-free transmission can be ensured. At the same time, we can use the asymmetric response of the reflection channel to achieve adjustable control of the light beam. For example, by randomly flipping the EP pairs, diffuse reflection and reciprocity-protected transmission⁹ can be achieved (**Fig. S10C**). **Figs. S10D** and **S10E** show the device when operating in reflection and transmission, respectively. Furthermore, it is possible to achieve different reflectivity in both directions by adjusting the proportion of the two symmetrical structures.

Our examples show that the concept of EP pairs extends beyond the reported polarization-control results presented in our manuscript. We have proposed other EP systems and explained how they could benefit from this EP superposition concept to solve problems unable to be coped with just one single EPs greatly expanding the applications of EP systems in general.”

Supplementary Figure 9. (A) Schematic of the unit cell. The structure is placed on a silica substrate with a ring and a square of silver resonators embedded in the silica space layer. The parameters are $p = 300$ nm, $L_1 = 175$ nm, $h_1 = 30$ nm, $R_{out} = 120$ nm, $R_{in} = 40$ nm, $h_2 = 20$ nm and $h_0 = 185$ nm. The incident wave is in the $+z$ or $-z$ direction. It is worth noting that the structure is polarization independent due to its inherent C_4 symmetry. (B) Spectral dependence of the matrix coefficients. An exceptional point,

where $r_{11} = 0$, is observed at $\lambda = 598$ nm. (C) Spectral dependence of matrix coefficients of the mirror symmetric structure. The other exceptional point in the EP pair, where $r_{22} = 0$, is observed at $\lambda = 598$ nm.

Supplementary Figure 10. Construction of EP Pairs for polarization independent systems. (A and B) Unidirectional reflectionless propagation phenomena corresponding to a pair of mirror symmetric structures, where the transmission matrix of (A) has $r_{11} = 0$, and (B) has $r_{22} = 0$. (C) Diffuse reflection and reciprocity-protected transmission achieved by randomly flipping a pair of EPs. The upper and lower figures show side and top views, respectively. (D and E) Visual effect demonstration in terms of reflection and transmission.

The above researches show that the concept of EP pair is widely applicable to other EP systems. Using a pair of symmetric EPs can break the inherent limitation of systems, which otherwise would respond to a single eigenstate only. As we propose, the general behavior of dual EP system is not only incremental technical advance, but also a conceptual innovation. We have also added the following description to the revised manuscript as, “It is worth noting that using pair of dual EPs is not only applicable to the above photonic systems, but can also be proven useful for the design of many other physical systems involving EPs, whether they display inherent chirality or not. (see more details in **Supplementary Note 2**)” Line 1-4, Page 7.

Comment 1.3.1: *Throughout the paper, the statements of novelty are jargon-heavy and quite narrow: "a general method to span the use of EP-induced singularities beyond the single degenerate eigenstate"; "combining PB-phase wave control of a pair of cross-polarized EPs eigenstates," "a general approach to extend the application of EPs to any arbitrary polarization state, surpassing thereby the inherent limitation of unique CP response of a system operating at an EP".*

Comment 1.3.2: *One of the specific new behaviors identified by the authors is that whereas previous metasurfaces "have been used to reconstruct fully polarized channels, but cross-polarized twin images were always projected at the opposite diffraction angle," their new metasurface can suppress such twin images.*

Such selling-points are scientifically valid, but I do not think they are of sufficiently broad interest.

Response 1.3.1: The authors thank the reviewer for this point. We have revised the statements of novelty in the revised manuscript as: “In this paper, we present a general method to span the effect of systems operating at EPs to any arbitrary polarization state, bypassing the intrinsic limitation of individual EPs, which operate on just one defined polarization state.” in Line 27-29, Page 3 and Line 1, Page 4.

and “Here we address this issue by endowing a pair of EPs on the two channels over which the PB phase operates, producing vectorial holographic images with arbitrary spatial polarization and free of undesired twin images.” in Line 17-20, Page 4.

Response 1.3.2: The authors thank the reviewer for this point. In the manuscript, we have revealed an idea for expanding the application dimension of EP, i.e., the combination of EP pairs. Subsequently, we used metasurface as the experimental platform to achieve the construction of arbitrary polarization states with EP protection, confirming our conjecture. As we mentioned in the **Abstract** and **Comment 1.2**, the construction of EP pairs is a general method that is not only applicable to the above system, but can also be proven useful for the design of many other physical systems involving EPs, whether they exhibit chirality or not. Actually, the “*One of the specific new behaviors identified by the authors is that whereas previous metasurfaces ‘have been used to reconstruct fully polarized channels, but cross-polarized twin images were always projected at the opposite diffraction angle,’ their new metasurface can suppress such twin images.*” you mentioned is only the response brought by Jones matrix EPs in our experimental system. In other EP systems, it can have different manifestations, such as unidirectional reflectionless propagation in a certain direction or vortex in a specific rotational direction. The removal of twin images in metasurfaces needs to be understood as an illustration of technological application made possible by our EP-pair idea.

As stated in **Comment 1.2**, the following description has been added to indicate that the approach of EP pair is also applicable to other EP systems:

“It is worth noting that using pair of dual EPs is not only applicable to the above photonic system, but can also be proven useful for the design of many other physical systems involving EPs, whether they display inherent chirality or not. (see more details in **Supplementary Note 2**)” in Line 1-4, Page 7.

Comment 1.4: *The paper does not present these issues to be particularly important either technologically (e.g., key bottlenecks for practical applications of metasurfaces; are the above twin images a critical limitation for anything?), or theoretical (e.g., capabilities that are fundamentally impossible to achieve via other design approaches). This are aspects required for high-impact general-purpose journals like Nature Communications.*

Response 1.4: Thanks for your critical comments, which have pushed us to illustrate the usefulness and applicability of pairs of EPs in other physical contexts, thereby improving the readability and impact of our work. As stated in **Comment 1.2**, this

approach can be extended to other systems operating at EPs to achieve the extension of asymmetric properties. Therefore, the construction of EP pairs can have universal applications.

We have highlighted the significance of the work in the manuscript as follows:

“And we believe that the idea of coupling a pair of EPs to realize the expansion of the asymmetric response can be applied to a wider range of EP systems.” in Line 4-6, Page 12.

As for technology, the suppression of the twin image can tear down some application barriers and achieve a wider light control. Twin images are generated from the opposite response of PB phase to left-handed and right-handed incident light, which is an inherent feature of simple PB engineered photonic system. This issue greatly limits the application of metasurfaces in various examples, such as:

- 1) Twin images will reduce the signal-to-noise ratio of the system. the twin image occupies a “potential” pixel in real space without adding new information, that is thus useless.
- 2) When the two images are close (or overlapped), crosstalk will occur, which will have an adverse effect on polarization measurement¹⁰. In addition, for image recognition, image reversal or even simple intensity reversal will cause cognitive obstacles¹¹ to the system, so when there are redundant images in the hologram, it is unfavorable for the subsequent intelligent recognition process.
- 3) The suppression of the conjugate property of PB phase can extend to many other applications. PB phase has been widely used in virtual reality or augmented reality¹². Due to the conjugate property of geometric phase, the metalens will perform different optical functions under different incident conditions (LCP or RCP). For example, an VR/AR device can only work in one polarization state. This is unfavorable for subsequent development, such as 3D display (which requires a pair of orthogonal polarization states). Our proposed design can solve this problem, avoid the difference between LCP and RCP modes, and realize the construction of full polarization state.

To sum up, our study proposes a generic method to solve the application obstacles caused by the asymmetric response of one EP. In addition, the removal of twin images is also significant for holography and related applications. We believe that these highlights can attract widespread interest and provide examples for future research work.

Comment 1.5: *Another issue with the presentation is that the connection between the physics of EPs and the metasurface-design strategy is quite hard to tease out. The problem starts with the second sentence of the abstract: "Because of their intrinsic chiral asymmetric responses, a given EP operates on only one and well-defined polarization channel, a specific property that enables intriguing asymmetric light control." Actually, this is NOT a general property of EPs, only a specific class of EPs in a certain narrow context (metasurfaces). EPs are mathematical constructs that don't have any intrinsic relation to physical concepts of "chirality" or "polarization"; even in other optics settings, EPs don't need to be related to polarization.*

Response 1.5: Thank you very much for your valuable suggestion. As you said, EPs

are mathematical constructs that have no intrinsic relation to chirality, and indeed there are also many EPs in non-chiral systems^{5,13,14} in previous studies. In our research, the plasmonic metasurface is selected as the experimental platform to prove our concept of EPs pair. I'm sorry that we didn't express this point more clearly in the second sentence of the abstract. What we wanted to express is that in the non-Hermitian system designed in the manuscript, due to the degeneracy of the eigenstates, the given EP only operates on one polarization channel. In order to better tease out the physical characteristics of EPs and the design strategy of metasurface, we have revised it as follows:

“In such systems, because of the degeneracy of the eigenstates, a given EP operates on only one and well-defined channel, a specific property that enables intriguing asymmetric control.” in Line 4-6, Page 2.

In addition, we took the referee comment in consideration to show that the approach is not only limited to specific optical systems. As such, we have revised the wording and propose a study case that do not bind EPs and polarization. As we mentioned in **Comment 1.2** and **Supplementary Note 2**, the idea of EP pairs also applies to non-chiral systems. In the second paragraph of the introduction, we have mentioned that metasurfaces are a platform for us to verify our theory, and it is precisely in the Jones matrix ($T = \begin{bmatrix} T_{xx} & T_{xy} \\ T_{yx} & T_{yy} \end{bmatrix}$) of the metasurfaces that T_{ij} is related to its polarization conversion ability, so the construction of full polarization state is of great significance for us to verify the role of the EPs pair. In fact, the idea of EPs pairs can be extended to more non-chiral EP systems.

Comment 1.6: *The first paragraph of the paper has a similar problem; it talks about both the general properties of EPs and the use of EPs in optics, but lacks a good logical flow from the general (mathematical) to the specific (light control). The paragraph also refers to concepts like EP handedness and EP order without proper introduction.*

Response 1.6: Thank you very much for your significant comment, which needs to be addressed clearly so as to accompany the reader and keep its interest in the rest of the article. We have reorganized the first paragraph of the manuscript to improve the logic of the article. In addition, EP handedness and EP order are also briefly introduced.

The discussion has been added in the revised manuscript as, “Such degeneracy of eigenstates has aroused extensive interest in the asymmetric control of the eigenstates by researchers in various fields, such as mechanics¹⁵, acoustics¹⁶, and electromagnetism¹⁷. Among them, optics is considered to be an ideal platform for observing the physical phenomena corresponding to EPs. Recent results led to chiral absorbers that can absorb perfectly only one particular state of the input wave¹⁸, chiral mode switching by encircling an EP¹⁹ or asymmetric transmission of circular polarized light²⁰. This asymmetry is thus endowed by the handedness of the EP (i.e., the system cannot coincide with its mirror image), which distinguishes between two opposite circularly-polarized beams. However, these efforts have always been limited to the exploitation of a single EP, often of second-order (i.e., EP in coupled-mode system, which can be expressed by a second-order matrix), restricting the response of the system to one single-polarization eigenstate defined by the handedness of the EP.” in

Comment 1.7: Later, in the "design methods" section, the role of the EPs does not stand out. The use of the EPs is embedded in a mass of details including the description of structural parameters, definition of the R matrix, and mirror symmetry analysis, which make the key ideas hard to stand out. I can basically follow what is going on, after multiple readings, but the presentation really ought to be improved.

Response 1.7: Thank you for your valuable suggestions. In order to make the key ideas more prominent, we have moved some details of the design method to the supplementary materials, and highlighted the description of EP in the main text.

1) We revised the diagram in **Fig. 1A** to **Fig. S1** and moved the following paragraphs into the supplementary materials:

“Supplementary Figure 1. Structure design of the metasurface. (A) Top view of one unit cell. **(B)** Side view of the metasurface. The topological vectorial metasurface consists of metal-dielectric-metal layers. The top metallic layer is composed of “L”-shaped meta-structure arrays (i.e., an L-shaped rod near-field coupled to a straight rod). It is followed by a SiO₂ spacer and an aluminum ground, which is employed as the metallic ground plane to block all the potential transmission channels, causing the metasurface to work in reflection. The period (P) of the unit cell is 300 nm to eliminate undesired diffraction effect in the visible. The main parameters of the “L”-shaped meta-structure (denoted as S) are L_1 , L_2 , g , w and L_3 . Here, $L_1 = 52$ nm, $L_2 = 140$ nm, $g = 70$ nm, $w = 50$ nm, $L_3 = 119$ nm, $h = 30$ nm, $h_1 = 40$ nm and $h_2 = 150$ nm.” in Line 2-11, Page 25.

Supplementary Figure 1. Structure design of the metasurface.

2) We replaced the original **Fig. 1A** with a new schematic diagram and made the

following modifications in the main text to make our description more concise:

“At the top, “└”-shaped meta-structure (denoted as S) arrays are used to construct the metallic device layer. The system is excited at normal incidence using a horizontally polarized light (LP-H) impinging on the device in the $-z$ direction. Among the different structural parameters of the meta-structure (Fig. S1), L_1 and L_3 are the two main parameters varied to achieve the desired EP response.” in Line 3-8, Page 6.

Figure 1. Design principle of topological vectorial metasurfaces with EPs pair. (A) Perspective view of the structural unit of topological vectorial metasurface. The horizontal polarized light is perpendicular to the metasurface along the $-z$ direction.

3) We have simplified the definition of R matrix and the analysis of mirror symmetry in the main text:

“To investigate the reflection matrix coefficients of the mirror-symmetric structure, mirror symmetric operator Π is applied to the reflection matrix \widehat{M} of given structure (Supplementary Note 1). Eq. S10 shows that under the condition of CP basis, the reflection matrix coefficients of \widehat{M} and \widehat{M}^m have a strict relationship of chiral inversion, which means that the polarization conversion capability of S structures on RCP light is exactly equal to that of S^m for LCP light.” in Line 17-23, Page 6.

We hope to have successfully addressed all the comments by reviewer #1 and, in particular, we hope to have presented clearly the interest and widespread potential of the original strategy introduced for the first time in this work.

Reply report to reviewer #2

Comment 2.1: *The manuscript reports the experimental demonstration of arbitrary polarization generation by using a pair of exceptional points (EPs). EPs are nanostructures that has non-zero conversion efficiency from LCP to RCP but zero from the opposite conversion (structure S). For a planar chiral structure, a mirror-symmetry one works opposite (structure S^m). Therefore, RCP component is purely converted from S and LCP component is purely converted from S^m with linearly polarized incidence. Arbitrary polarization and holographic images can be generated by multiplexing structures S and S^m and PB phase.*

Holographic images with different polarization and vectorial wave front shaping are demonstrated with EPs. The manuscript demonstrates novel results. The theory in Supplementary Information well supports their design and experimental results. Overall, the manuscript can be accepted for publication after a minor revision considering the following comments.

Response 2.1: We are grateful to the reviewer's positive feedback and recommendation to publish in *Nature Communications*.

Comment 2.2: *There are too many symbols, and they might make readers confused. For example:*

a. *Azimuth and ellipticity angles: They repeat in the whole manuscript. The authors should note the symbol the first time mentioned them (lines 97 and 98).*

b. *Symbol for azimuth angle is φ (psi) in line 248 but is ϕ (phi) in line 447.*

c. *There are too many ϕ s (e.g., orientation angle ϕ , initial orientation angle ϕ_{\pm} , angle increment ϕ_d) and might not be necessary. I suggest the authors not use symbols for those only mention once or twice.*

d. *Every time I read $|-\rangle$ and $|+\rangle$, I cannot link the results with RCP and LCP by intuition. I suggest the authors replace them with $|R\rangle$ and $|L\rangle$.*

e. *The matrix R can be replaced by M (refers to "m"etasurface or "m"atrix) so that it would not be confused with RCP.*

Response 2.2: We are grateful to the reviewer for its pertinent comments on our manuscript. We have realized that there are too many symbols in the manuscript to affect readers' understanding, so we have made the following modifications:

a. We have marked the corresponding symbols when we first mentioned azimuth and ellipticity angles, and the modifications are as follows:

"Moreover, by decoupling the intensity and polarization information, we utilize the plasmonic non-Hermitian metasurface to independently encode the azimuth (\$\varphi\$ ) and ellipticity (\$\chi\$ ) angles of the polarization in a uniformly distributed intensity profile." in Line 26-29, Page 4.

b. Sorry for your doubts caused by our negligence. We have used \$\varphi\$ (psi) to

represent all azimuth angles, and the corresponding changes are as follows:

“(E) LP-45° ($\varphi = 45^\circ, \chi = 0^\circ$) and (F) ELP ($\varphi = 0^\circ, \chi = 18.43^\circ$).” in Line 8, Page 17.

c. We have carefully reviewed all the symbols related to φ , and deleted φ_+ , φ_- and φ_{\pm} with fewer occurrences. As for orientation angle φ and angle gradient φ_d , since they appear more often and need to appear in the formula, we have decided to keep them in the manuscript. Relevant changes are as follows:

“for LCP incident light we arrange the S meta-structures on a horizontal line and rotate them clockwise with the set initial orientation angle and an angle increment of φ_d for each adjacent meta-structures, i.e., to deflect the resulting RCP output light to the right side with a deflection angle of θ_t . Similarly, for RCP incident light, by rotating the S^m meta-structures counterclockwise with another initial orientation angle and setting the same angle increment φ_d , the output LCP can achieve the same deflection angle. By encoding the holographic phase information on both LCP and RCP beams starting from the initial orientation angle and applying a modified Gerchberg–Saxton (GS) algorithm,” in Line 8-16, Page 8.

d. Thanks for your valuable suggestion. We have replaced all the $|-\rangle$ s and $|+\rangle$ s with $|R\rangle$ s and $|L\rangle$ s respectively, which does make the text clearer. Some of the changes are shown below as examples:

“For each of the mirror-symmetric meta-structure S^m , the current distribution of the vertical arm switches the flow direction with respect to those in S as shown in Fig. 1C, resulting in the flipping of the degeneracy from $E_x - iE_y$ to $E_x + iE_y$, i.e., from RCP to LCP (denoted as $|R\rangle$ and $|L\rangle$, respectively).” in Line 14-17, Page 6.

e. Thank you very much for your suggestion. We have denoted the matrix with M instead of R to distinguish it from RCP. And some of the changes are shown below as examples:

“To investigate the reflection matrix coefficients of the mirror-symmetric structure, mirror symmetric operator Π is applied to the reflection matrix \widehat{M} of given structure (Supplementary Note 1). Eq. S10 shows that under the condition of CP basis, the reflection matrix coefficients of \widehat{M} and \widehat{M}^m have a strict relationship of chiral inversion, which means that the polarization conversion capability of S structures on RCP light is exactly equal to that of S^m for LCP light.” in Line 17-23, Page 6.

Comment 2.3: *Does the initial orientation angle for S and S^m affect the polarization state in the far field (e.g., results in Figs 1I, 3E and 3F) if multiplexing a pair of EPs or more?*

Response 2.3: Thank you very much for your insightful question. In our manuscript, the role of structure S is to convert LCP to RCP and assign a PB phase, while structure S^m converts RCP to LCP and assigns another PB phase in the opposite phase sign. The other channel of S and S^m (i.e., RCP incidence for S and LCP for S^m), which is not endowed with a PB phase, will exit vertically along the incident light path and will not participate in the holographic imaging. Briefly, for S , the following relationship exists:

$$|L\rangle = e^{i2\varphi_R}|R\rangle \quad (2-1)$$

Also, for S^m , there exists:

$$|R\rangle = e^{-i2\varphi_L}|L\rangle \quad (2-2)$$

where φ_R and φ_L represent the initial orientation angles of S and S^m . That is, when only one type of structure is used, the polarization state of the output beam will be one of two circular polarization states with a specific phase attached.

When a pair of EPs is multiplexed (as shown in **Fig. 3E**), the outgoing beam will be a combination of RCP and LCP, which can be expressed by the following equation:

$$|n\rangle = A_R e^{i2\varphi_R}|R\rangle + A_L e^{i2\varphi_L}|L\rangle \quad (2-3)$$

where A_R and A_L are the amplitude of the RCP and LCP beams, respectively. The azimuth angle ψ and ellipticity angle χ of $|n\rangle$ can be calculated by $\psi = \varphi_R - \varphi_L$ and $\chi = \frac{1}{2} \arcsin \frac{A_R^2 - A_L^2}{A_R^2 + A_L^2}$. In the above case, due to the equal number of structures S and S^m , and due to their equal scattering response, the values of A_R and A_L are equal. Therefore, the output beam will be coupled as linearly polarized light with azimuthal angle $\psi = \varphi_R - \varphi_L$ and ellipticity $\chi = 0$. It can be found that the initial orientation angle will affect the magnitude of the azimuth angle, thereby changing the polarization state.

When the number of S and S^m is not equal, the situation becomes more complicated. Take the simplest case as an example, that is, coupling two S and one S^m (as shown in **Fig. 3F**) elements, the value of A_R can be adjusted by the following formula:

$$A_R = \sqrt{(1 + \cos 2\Delta\delta_R)} \quad (2-4)$$

where $\Delta\delta_R$ represents the rotation angle difference of the two structures S . It indicates that the difference in the initial orientation angles of the two structures S has an impact on the ellipticity angle. In addition, as mentioned earlier, the azimuth angle is also related to the initial rotation angles of S and S^m . Briefly speaking, in this case, the initial orientation angles for S and S^m will affect the polarization state in the far field.

In summary, when multiplexing a pair of EPs or more, the initial orientation angles for S and S^m will always affect the polarization state in the far field.

Comment 2.4: *Figure captions of Figs 3E and 3F shows the corresponding azimuth and ellipticity angles. However, there is no mention in the main text. Could the author explain more about how to design specific polarization of output holographic images?*

Response 2.4: Thanks for your important advice. In the previous part of the manuscript, we have confirmed that the structure shown in **Fig. 4A** can independently distribute arbitrary phase information in the metasurface plane to each pixel, which provides the basis for the control of far-field amplitude and polarization information.

To decouple amplitude from far-field polarization information, we use a modified Gerchberg-Saxton (GS) algorithm for vectorial fields. Assuming that the intensity of the far field is I^f , the azimuth angle is ψ^f , and the ellipticity angle is χ^f (superscript f represents far-field image plane), we can obtain the amplitude information of the LCP

(α_L^f) and RCP (α_R^f) in the far field:

$$\alpha_L^f = \sqrt{[I^f - I^f \sin(2\chi^f)]/2} \quad (2-5)$$

$$\alpha_R^f = \sqrt{[I^f + I^f \sin(2\chi^f)]/2} \quad (2-6)$$

$$\alpha^f = 2\psi^f \quad (2-7)$$

where α^f represent the phase difference between LCP and RCP. Subsequently, a random phase φ_{rd} is given to the amplitude information described above to obtain the initial complex amplitude $(\alpha_{L,R}^f e^{i\varphi_{rd}})$, and then the initial metasurface information is obtained by the inverse Fourier transform:

$$B_L^m(1) = \mathcal{F}^{-1}(\alpha_L^f e^{i\varphi_{rd}}) \quad (2-8)$$

$$B_R^m(1) = \mathcal{F}^{-1}(\alpha_R^f e^{i\varphi_{rd}}) \quad (2-9)$$

Afterwards, the iterative Fourier transform process with the iteration number j from 1 to N is applied. If j is an odd number, the algorithm is described as:

$$\left\{ \begin{array}{l} C_R^f(j) = \mathcal{F}(e^{i\angle[B_R^m(j)]}) \\ B_R^m(j+1) = \mathcal{F}^{-1}(\alpha_R^f e^{i\angle[C_R^f(j)]}) \\ C_L^f(j) = \mathcal{F}(e^{i\angle[B_L^m(j)]}) \\ B_L^m(j+1) = \mathcal{F}^{-1}(\alpha_L^f e^{i(\angle[C_L^f(j)] - \alpha^f)}) \end{array} \right. \quad (2-10)$$

And if j is an even number, the algorithm becomes:

$$\left\{ \begin{array}{l} C_L^f(j) = \mathcal{F}(e^{i\angle[B_L^m(j)]}) \\ B_L^m(j+1) = \mathcal{F}^{-1}(\alpha_L^f e^{i\angle[C_L^f(j)]}) \\ C_R^f(j) = \mathcal{F}(e^{i\angle[B_R^m(j)]}) \\ B_R^m(j+1) = \mathcal{F}^{-1}(\alpha_R^f e^{i(\angle[C_R^f(j)] + \alpha^f)}) \end{array} \right. \quad (2-11)$$

where $\angle[D]$ represent the phase of D . When the iteration process ends, the final phase information of the metasurface is as follows:

$$\varphi_L^m = \angle[B_L^m(j)] \quad (2-12)$$

$$\varphi_R^m = \angle[B_R^m(j)] \quad (2-13)$$

Considering the relationship between the rotation angle of the meta-structure and the geometric phase, the rotation angle is determined as:

$$\Delta_L = -\frac{1}{2}\varphi_L^m \quad (2-14)$$

$$\Delta_R = \frac{1}{2}\varphi_R^m \quad (2-15)$$

Therefore, we can encode two different images as azimuth and ellipticity angles into a uniformly distributed intensity profile to achieve arbitrary combination of spatial variation amplitude and polarization information.

It is worth noting that since the number of rows of S and S^m in the metasurface is equal, there is no additional degree of freedom to control the intensity difference between LCP and RCP. We need to tune the orientation and ellipticity (translation and scaling) to make the total strength of LCP and RCP equal in the far field.

The corresponding changes to explain this situation were made in the manuscript as follows:

“The modified Gerchberg-Saxton (GS) algorithm of vector field is used to realize the decoupling of amplitude and far-field polarization information. It is noteworthy that because the number of rows of S and S^m in the metasurface is equal, there is no additional degree of freedom to control the intensity difference between LCP and RCP. As compensation, the azimuth and ellipticity angles need to be tuned (translated and scaled) to make the total strength of LCP and RCP equal in the far field.” in Line 25-29, Page 10 and Line 1-2, Page 11.

Comment 2.5: *There are no explanation of the abbreviation “Es1” and “Es2” shown in Fig 2c.*

Response 2.5: We are sorry for the confusion caused by our oversight. The abbreviation “Es1” and “Es2” shown in **Fig 2C** stand for eigenstate1 and eigenstate 2, respectively. Their projections on the Poincaré sphere are functions of wavelength and degenerate at the two poles (RCP and LCP) at $\lambda = 600$ nm. The corresponding changes were made in the manuscript as follows:

“(B and C) The eigenstates of meta-structures S (B) and S^m (C) projected on the Poincaré sphere as a function of wavelength, showing that they degenerate at the two poles (RCP and LCP) at $\lambda = 600$ nm, where Es1 and Es2 represent eigenstate 1 and eigenstate 2.” in Line 21-25, Page 16.

Comments 2.6: *The authors demonstrate 2 eigenstates degenerate at RCP and LCP. Is there any limitation to degenerate 2 eigenstates at arbitrary elliptical polarization?*

Response 2.6: Thank you very much for your significant and insightful comment. In the manuscript, the structure we chose to adopt can be approximated as a planar structure. The plane of the structure is set as the xy plane, and the beam is perpendicular to the structure surface along the $-z$ direction.

Under the condition of coherent monochromatic plane wave, the generalized Jones calculus is used to connect the complex amplitudes of the incident field and the reflected field:

Fig. R2-1. Schematic of the geometry. The meta-atoms are located in the xy plane with light impinging normally to the structure in the z direction. The black dashed lines indicate the mirror plane.

$$\begin{pmatrix} r_x \\ r_y \end{pmatrix} = \begin{pmatrix} M_{xx} & M_{xy} \\ M_{yx} & M_{yy} \end{pmatrix} \begin{pmatrix} i_x \\ i_y \end{pmatrix} = M^f \begin{pmatrix} i_x \\ i_y \end{pmatrix} \quad (2-16)$$

For convenience, M_{ij} can be replaced by the letters A-D, i.e.

$$\begin{pmatrix} r_x \\ r_y \end{pmatrix} = \begin{pmatrix} A & B \\ C & D \end{pmatrix} \begin{pmatrix} i_x \\ i_y \end{pmatrix} \quad (2-17)$$

In general, if metamaterials do not exhibit any reflection or rotational symmetry, then all complex components of the Jones matrix are different. However, when there is mirror symmetry with respect to the plane perpendicular to the z axis, the reflected structure of the meta-atom about the yz plane is the same as that seen from the back:

$$E_x^{-1} M^f E_x = \begin{pmatrix} A & -B \\ -C & D \end{pmatrix} = M^b \quad (2-18)$$

where E_x is the reflection matrix with respect to the x axis, and the superscript f or b indicates that the viewing direction is forward or backward.

When only reciprocal media are considered, we can get²¹:

$$M^b = \begin{pmatrix} A & -C \\ -B & D \end{pmatrix} \quad (2-19)$$

Then there is the following relationship:

$$\begin{pmatrix} A & -B \\ -C & D \end{pmatrix} = \begin{pmatrix} A & -C \\ -B & D \end{pmatrix} \quad (2-20)$$

And we can get $B = C$, i.e.,

$$M^f = \begin{pmatrix} A & B \\ B & D \end{pmatrix} \quad (2-21)$$

The results show that the non-diagonal elements of the corresponding Jones matrix

are the same when the beam is perpendicular to the metasurface with z -axis mirror symmetry. And the degenerate state corresponding to this kind of matrix can only be LCP or RCP.

If this binding is broken and elliptical polarization state is taken as the degenerate state, the required Jones matrix should be in the following form:

$$M = \begin{pmatrix} A & B \\ C & D \end{pmatrix} \quad (2-22)$$

This requires introducing the asymmetry in the z direction (as shown in **Fig. R2-2**), or changing the incident condition from perpendicular to oblique. Regrettably, there is no report on EP that can accurately address any polarization state at present. For information, we are currently working on this subject and we can prove that non normal incidence is a valuable solution to realize arbitrary EP polarization. We believe that these results extend beyond the scope of the manuscript. The referee question is nevertheless relevant and deserves dedicated explanation and demonstration, which will be provided in forthcoming work from the team.

Fig. R2-2. Schematic of the geometry. A three-dimensional structure made of an L -shaped structure and an I -shaped structure with no symmetry at all.

Comments 2.7: *In lines 227-229, the authors address that “The holographic images are set to 30° by controlling the orientation angle increment $\phi_d = 45^\circ$.” Could the authors explain more about the relation between orientation angle increment and the angle the holographic images reconstructed.*

Response 2.7: Thank you very much for your significant comment. The relationship between orientation angle increment and the angle of holographic image is obtained from the generalized laws of reflection and refraction derived from Fermat's principle. Yu et al.²² showed that anomalous reflection and refraction can be observed at the interface with constant phase gradient. The EM response of the metasurface to different polarized transmission waves is independent. Therefore, the generalized refraction law can be written as:

$$\begin{cases} \sin(\theta_t) n_{t\parallel} - \sin(\theta_i) n_{i\parallel} = \frac{\lambda_0}{2\pi} \frac{d\varphi_x}{dx} \\ \sin(\theta_t) n_{t\perp} - \sin(\theta_i) n_{i\perp} = \frac{\lambda_0}{2\pi} \frac{d\varphi_y}{dy} \end{cases} \quad (2-23)$$

where θ_i is the incident angle of the EM wave, θ_t is the refraction angle, φ_x and φ_y represent the transmission phase in x and y directions respectively, n_i and n_t represent the refractive index of the two media, subscript \parallel and \perp represent horizontal polarization wave and vertical polarization wave respectively.

Assume that the transmitted wave propagates along the z-axis, that is, $\theta_i = 0$, the refraction angle of horizontal and vertical polarization can be calculated as:

$$\begin{cases} \theta_{t\parallel} = \arcsin\left(\frac{\lambda_0}{2\pi} \frac{d\varphi_x}{dx}\right) \\ \theta_{t\perp} = \arcsin\left(\frac{\lambda_0}{2\pi} \frac{d\varphi_y}{dy}\right) \end{cases} \quad (2-24)$$

In the research related to metasurfaces, there is often only one dimension that needs to be considered, so $\frac{\Delta\varphi}{p}$ can be used instead of $\frac{d\varphi_x}{dx}$ or $\frac{d\varphi_y}{dy}$, and the formula can be simplified as:

$$\theta_t = \arcsin\left(\frac{1}{k_0} \frac{\Delta\varphi}{p}\right) \quad (2-25)$$

where k_0 is the wavenumber in the free space and p is the period of the unit-cell.

According to the Pancharatnam–Berry phase, the difference of rotation angles of adjacent structures has the following relationship with the resulting phase difference:

$$\Delta\varphi = 2 * \varphi_d \quad (2-26)$$

Finally, the relationship between orientation angle increment and the angle the holographic images reconstructed can be described as:

$$\theta_t = \arcsin\left(\frac{2}{k_0} \frac{\varphi_d}{p}\right) \quad (2-27)$$

Therefore, when $\varphi_d = 45$ degrees, the deflection angle θ_t is 30 degrees.

The corresponding changes were made in the manuscript as follows:

“According to the formula:

$$\theta_t = \arcsin\left(\frac{2\varphi_d}{k_0 p}\right) \quad (1)$$

The holographic images are set to 30° by controlling the orientation angle increment $\varphi_d = 45^\circ$ (where k_0 is the wavenumber in the free space and p is the period of the unit-cell).” in Line 27-29, Page 9 and Line 1-3, Page 10.

Comments 2.8: *In lines 253-255, the authors address that ‘the images of “Infinity” (Fig. S4B) and “Tai Chi” (Fig. S4C) represent the azimuth angle and ellipticity angle, respectively.’ Does the “intensity distribution” of the reconstructed images represent the value of spatial distribution of azimuth/ellipticity angle? What are the values of azimuth/ellipticity angle we can be observed from the image?*

Response 2.8: Thank you very much for your significant comment. In **Figs. S5B** and **S5C**, the intensity of the images represents the spatial distribution value of the azimuth/ellipticity angle, respectively, with the azimuth angle ranging from 0° to 180° and the

ellipticity angle ranging from -45° to 45° . To make our description easier to understand, we have added scale bars to **Fig. S5**.

Supplementary Figure 5. Design of the spatially distributed polarization profile. Simulated vectorial hologram with distribution of (A) intensity, (B) azimuth and (C) ellipticity angles of the polarization in the far-field.

As for the "intensity distribution" mentioned in **Fig. S5A**, it is not directly related to the azimuth and ellipticity angles. Actually, the three pictures are relatively independent, they only need to ensure that the patterns of azimuth and ellipticity angles are in the region of intensity distribution. In this manuscript, we choose a uniformly distributed circular spot as the intensity distribution for convenience. And "Infinity" and "Tai Chi" are two images arbitrarily constructed in this area.

It should be added that the azimuth and ellipticity angles are achieved through subsequent calculations based on the measured experimental data, and cannot be directly observed from experiments. The decoding of the polarization information is based on the Stokes parameters, which can be obtained by introducing a quarter-wave plate and a linear polarizer in the optical setup. The intensity of the optical beam after the waveplate and linear polarizer can be expressed as²³⁻²⁵:

$$I(\theta, \phi) = \frac{1}{2}(S_0 + S_1 \cos 2\theta + S_2 \sin 2\theta \cos \phi - S_3 \sin 2\theta \sin \phi) \quad (2-28)$$

where θ is the rotation angle of the linear polarizer and ϕ is the phase of the waveplate. By measuring four intensity values, $I(0^\circ, 0^\circ)$, $I(45^\circ, 0^\circ)$, $I(90^\circ, 0^\circ)$ and $I(45^\circ, 90^\circ)$, the Stokes parameters are obtained:

$$\begin{cases} S_0 = I(0^\circ, 0^\circ) + I(90^\circ, 0^\circ) \\ S_1 = I(0^\circ, 0^\circ) - I(90^\circ, 0^\circ) \\ S_2 = 2I(45^\circ, 0^\circ) - I(0^\circ, 0^\circ) - I(90^\circ, 0^\circ) \\ S_3 = I(0^\circ, 0^\circ) + I(90^\circ, 0^\circ) - 2I(45^\circ, 90^\circ) \end{cases} \quad (2-29)$$

And the azimuth angle and ellipticity angle can be extracted as:

$$\psi = \frac{1}{2} \tan^{-1} \left(\frac{S_2}{S_1} \right) \quad (2-30)$$

$$\chi = \frac{1}{2} \sin^{-1} \left(\frac{S_3}{S_0} \right) \quad (2-31)$$

Comments 2.9: Both images of "Infinity" and "Tai Chi" are binary (either 1 or 0).

Could the authors achieve arbitrary combinations (e.g., grayscale images) of ellipticity and azimuth angles in the selected region, as mentioned in line 250-251?

Response 2.9: Thank you very much for your valuable suggestion. As the referee correctly indicates, in **Fig. 4** and **Fig. S5** we use binary images to show arbitrary combination of azimuth and ellipticity angles. In fact, this Modified GS algorithm can adapt to more complex situations, such as grayscale images. The answer to the referee question regarding the grayscale projection is: yes, it is possible, and we have provided an additional demonstration as requested to support the claim in line 250-251. **Fig. R2-3** shows the asymmetric vectorial meta-hologram realized by grayscale images, in which the intensity, azimuth and ellipticity angles are all replaced by grayscale images. In **Fig. R2-3A**, the grayscale image of Van Gogh's self portrait is used as intensity. **Fig. R2-3B** shows Van Gogh's "The Starry Night", and **Fig. R2-3C** is a grayscale image of "Rest from Work". The corresponding simulation results show that the meta-holograms are well reconstructed, confirming the potential of the Modified GS algorithm, which can achieve arbitrary combination of ellipticity and azimuth angles in the selected region.

Fig. R2-3. Design of the spatially distributed polarization profile. (A to C) Designed vectorial hologram with the distribution of (A) intensity, (B) azimuth and (C) ellipse angle. (D to F) Simulated vectorial hologram with distribution of (A) intensity, (B) azimuth and (C) ellipticity angles of the polarization in the far-field.

Comments 2.10: *There are some famous literatures about demonstration of independent phase profile for RCP and LCP (or arbitrary orthogonal elliptical polarizations) incident light [Nat. Nanotech. 10, 937-943 (2015) & Phys. Rev. Lett. 118, 113901 (2017)]. They utilize both propagation phase and PB phase of nanostructures in the design. Therefore, there is no necessary to have orthogonal EP eigenstates pairs in the design. Could the authors compare the difference and address their advantages?*

Response 2.10: Thank you very much for your significant comment. The two literature articles mentioned by the referee are very famous and provided a lot of inspiration. In the first one, Arbabi et al.²⁶ used a uniform birefringent metasurface to achieve arbitrary symmetry and unitary Jones matrix; they prepared a high-contrast dielectric elliptical nano-pillar metasurface platform, which provides complete control of polarization and phase. At the same time, this platform has a high average transmission rate (more than 85%), which is of great significance for practical applications. In the second work, Mueller et al.²⁷ showed that the geometric phase and propagation phase used in tandem allow arbitrary phase profile to be applied on any two orthogonal polarization states (linear, circular or elliptical). In these two works, the researchers adjusted the size and rotation angle of the nano-pillars, and carried out the required phase and polarization changes of the transmitted light by combining the propagation phase and the PB phase to generate the required spatially varying polarization and phase distribution.

As such, and in all existing works, PB phase occurs on both conversion channels equally but with an opposite phase sign. It appears that in our system, the PB phase only operates on one of the two polarization conversion channels, simply because the other polarization channel vanishes at the EP. That is, only one polarization state is affected by rotating the structure, while the other state remains unaffected (because it is always zero), which is somehow different from conventional mixing of two non-zero phase modulation processes. Taking advantage of our EP-induced asymmetric phase response, we realized asymmetric vectorial wavefront modulation and achieved the construction of arbitrary polarization state. In the above literatures, due to the wavelength sensitivity of the propagation phase, the mode of PB+propagation phase can only operate under specific wavelength conditions, making it difficult to achieve broadband applications. However, EPs can maintain the asymmetric response over a wide wavelength range (as shown in **Fig. 2A**). Combined with the inherent broadband performance of the PB phase, the method we provided can achieve wavefront control under broadband conditions.

To highlight the characteristics of our work, corresponding descriptions have been added to the manuscript as follows:

“It should be noted that the asymmetric vectorial hologram can be designed to provide broadband response since the channels M_{RL} and M_{LR}^m are always larger than M_{LR} and M_{RL}^m in a wide frequency range as shown in **Fig. 2A**” in Line 12-14, Page 11.

In summary, we have addressed all the comments of reviewer #2.

References:

- 1 Tong, Q., Li, J. & Wang, S. Acoustic circular dichroism in a three-dimensional chiral metamaterial. *arXiv preprint arXiv:2301.02526* (2023).
- 2 Park, S. H. et al. Observation of an exceptional point in a non-Hermitian metasurface. *Nanophotonics* **9**, 1031-1039 (2020).
- 3 Gao, T. et al. Chiral modes at exceptional points in exciton-polariton quantum fluids. *Phys. Rev. Lett.* **120**, 065301 (2018).
- 4 Liu, T. et al. Single-sided acoustic beam splitting based on parity-time symmetry. *Phys. Rev. B* **102**, 014306 (2020).
- 5 Gu, X. et al. Unidirectional reflectionless propagation in a non-ideal parity-time metasurface based on far field coupling. *Opt. Express* **25**, 11778-11787 (2017).
- 6 Dong, S. et al. Loss-assisted metasurface at an exceptional point. *Acs Photonics* **7**, 3321-3327 (2020).
- 7 Jin, X. R., Zhang, Y. Q., Zhang, S., Lee, Y. & Rhee, J. Y. Polarization-independent electromagnetically induced transparency-like effects in stacked metamaterials based on Fabry-Perot resonance. *J. Optics-UK* **15**, 125104 (2013).
- 8 Chen, J., Wang, C., Zhang, R. & Xiao, J. Multiple plasmon-induced transparencies in coupled-resonator systems. *Opt. Lett.* **37**, 5133-5135 (2012).
- 9 Chu, H. et al. Diffuse reflection and reciprocity-protected transmission via a random-flip metasurface. *Sci. Adv.* **7**, eabj0935 (2021).
- 10 Zhang, X. et al. Direct polarization measurement using a multiplexed Pancharatnam-Berry metahologram. *Optica* **6**, 1190-1198 (2019).
- 11 Tai, L., Li, S. & Liu, M. A deep-network solution towards model-less obstacle avoidance. *IROS IEEE*, 2759-2764 (2016).
- 12 Lee, G.-Y. et al. Metasurface eyepiece for augmented reality. *Nat. Commun.* **9**, 4562 (2018).
- 13 Chong, Y. D., Ge, L. & Stone, A. D. PT-Symmetry Breaking and Laser-Absorber Modes in Optical Scattering Systems. *Phys. Rev. Lett.* **106**, 093902 (2011).
- 14 Dey, S., Laha, A. & Ghosh, S. Nonlinearity-induced anomalous mode collapse and nonchiral asymmetric mode switching around multiple exceptional points. *Phys. Rev. B* **101**, 125432 (2020).
- 15 Bender, C. M., Berntson, B. K., Parker, D. & Samuel, E. Observation of PT phase transition in a simple mechanical system. *Am. J. Phys.* **81**, 173-179 (2013).
- 16 Fleury, R., Sounas, D. & Alù, A. An invisible acoustic sensor based on parity-time symmetry. *Nat. Commun.* **6**, 5905 (2015).
- 17 Bittner, S. et al. P t symmetry and spontaneous symmetry breaking in a microwave billiard. *Phys. Rev. Lett.* **108**, 024101 (2012).
- 18 Sweeney, W. R., Hsu, C. W., Rotter, S. & Stone, A. D. Perfectly Absorbing Exceptional Points and Chiral Absorbers. *Phys. Rev. Lett.* **122**, 093901 (2019).
- 19 Li, A. et al. Hamiltonian Hopping for Efficient Chiral Mode Switching in Encircling Exceptional Points. *Phys. Rev. Lett.* **125**, 187403 (2020).
- 20 Lawrence, M. et al. Manifestation of PT Symmetry Breaking in Polarization Space with Terahertz Metasurfaces. *Phys. Rev. Lett.* **113**, 093901 (2014).
- 21 Menzel, C., Rockstuhl, C. & Lederer, F. Advanced Jones calculus for the classification of

- periodic metamaterials. *Phys. Rev. A* **82**, 053811 (2010).
- 22 Yu, N. et al. Light Propagation with Phase Discontinuities: Generalized Laws of Reflection and Refraction. *Science* **334**, 333-337 (2011).
- 23 Song, Q. et al. Broadband decoupling of intensity and polarization with vectorial Fourier metasurfaces. *Nat. Commun.* **12**, 3631 (2021).
- 24 Ferrand, P., Allain, M. & Chamard, V. Ptychography in anisotropic media. *Opt. Lett.* **40**, 5144-5147 (2015).
- 25 Vyas, S., Kozawa, Y. & Sato, S. Polarization singularities in superposition of vector beams. *Opt. Express* **21**, 8972-8986 (2013).
- 26 Arbabi, A., Horie, Y., Bagheri, M. & Faraon, A. Dielectric metasurfaces for complete control of phase and polarization with subwavelength spatial resolution and high transmission. *Nat. Nanotechnol.* **10**, 937-943 (2015).
- 27 Mueller, J. B., Rubin, N. A., Devlin, R. C., Groever, B. & Capasso, F. Metasurface polarization optics: independent phase control of arbitrary orthogonal states of polarization. *Phys. Rev. Lett.* **118**, 113901 (2017).

REVIEWER COMMENTS

Reviewer #1 (Remarks to the Author):

The revised paper is significantly clearer about the novelty of this work. In my view, the paper could be published in Nature Communications, if the following issues are resolved. Most of these are presentation/explanation issues involving the first few pages of the paper.

- The abstract should be even more specific about what the demonstration structure achieved. Right now it says "arbitrary, yet unidirectional, vectorial wave front shaping", but this is not specific enough.

- The first paragraph goes out of its way to point out that in previous schemes for exploiting EPs, the EPs are "often of second-order". This seems to be implying that a noteworthy aspect of this work will be the generalization to higher-order EPs, but it seems that the point is subsequently dropped without any further discussion. From my understanding, the authors are using two simultaneous EPs, but these behave as a pair of decoupled second-order EPs, not a higher-order EP. So why point out the "second-order" aspect? If it's a tangential point related to future generalizations, it is best moved to the conclusion.

- Line 83: "reflection/transmission zero singularities" -- the use of "reflection/transmission" is ambiguous about whether the zero occurs in the reflection AND transmission simultaneously, or reflection OR transmission. For clarity, this passage should be immediately followed up by a quick description of what these singularities mean, with minimal jargon.

- In the text accompanying the description of Fig. 1a, L1 and L3 are suddenly referred to, but the reader only finds out what these mean by looking at Fig. 1d. The subplots should be rearranged so that the order follows the logic in the text discussion.

- Line 125: "the degeneracy" is referred to without proper introduction/discussion of what is degenerate.

- Fig. 1i is only vaguely described in the text and the caption. This is not good, because it's directly related to the main selling point of "engineering arbitrary states", whereas the previous subplots, a-h, are more about controlling one polarization state (similar to earlier papers).

- The graphics in Fig. 1i itself are rather small and sparsely labelled. This is surprising since this is the main design idea for the experimental device! The key point to convey here is that there are two types of meta-atoms (targeting different EPs related by mirror symmetry) laid out on a single metasurface, with rotations to induce a PB phase in each channel.

Reviewer #2 (Remarks to the Author):

The authors have addressed the technical issues I raised in my previous review. The experimental results clearly align with the theoretical calculations and the design.

In Comment 1.2, Reviewer #1 expressed doubt about the incremental technical advancement of the metasurface design using EPs. However, after reviewing previous works on polarization control using EPs, as well as the theories presented in the Supplementary Information, I am convinced that this manuscript offers a novel design principle and advanced technology that overcomes previous limitations.

In Comment 1.3.2, Reviewer #1 expressed concern about whether this work would be of interest to a broader readership. However, I believe that Supplementary Note 2 provides sufficient information on the concept of EP pairs, which would be of interest to readers for those studying EP systems. In my opinion, the removal of twin images in metasurfaces is a significant challenge, and this work should be of broad interest to the metasurface community.

Overall, I highly recommend this manuscript for publication in Nature Communications. However, I do have a few suggestions for the authors.

1. It would be of interest to some readers to understand how the circular-polarization-based EP can be extended to achieve full-polarization reconstruction. I recommend including the statement from Response 2.3 in the Supplementary Information.

2. Many researchers studying vectorial holography may have the same question addressed in Comment 2.4. Therefore, I recommend including the statement from Response 2.4 and the simulated gray-scale vectorial hologram (Fig. R2-3) in the Supplementary Information. This would provide valuable additional information for readers interested in vectorial holography.

Reply report to reviewer #1

Comment 1.1: *The revised paper is significantly clearer about the novelty of this work. In my view, the paper could be published in Nature Communications, if the following issues are resolved. Most of these are presentation/explanation issues involving the first few pages of the paper.*

Response 1.1: We appreciate the reviewer's positive evaluation and his/her understanding of the novelty, which make the referee recommend to publish in *Nature Communications*.

Comment 1.2: *The abstract should be even more specific about what the demonstration structure achieved. Right now it says "arbitrary, yet unidirectional, vectorial wave front shaping", but this is not specific enough.*

Response 1.2: Thank you very much for your valuable suggestion. In the abstract, arbitrary, unidirectional and vectorial wave front shaping refer respectively to the realization of arbitrary polarization states, the suppression of redundant images due to the conjugation effect of the PB phase and the implementation of arbitrary design of the polarization information (azimuth and ellipticity angles) in the far-field. To make our statement more specific, or rather clearer, the corresponding changes were made in the manuscript as follows:

“Non-Hermitian reflective PB metasurfaces designed using such EP superposition enable arbitrary, yet unidirectional, vectorial wave front shaping devices. As a proof of principle, we demonstrate arbitrary polarized metasurface holography that effectively suppresses the redundant twin images generally projected due to the conjugated PB phase occurring on the two crossed circular-polarized light beams.” in Line 21-24, Page 2.

Comment 1.3: *The first paragraph goes out of its way to point out that in previous schemes for exploiting EPs, the EPs are "often of second-order". This seems to be implying that a noteworthy aspect of this work will be the generalization to higher-order EPs, but it seems that the point is subsequently dropped without any further discussion. From my understanding, the authors are using two simultaneous EPs, but these behave as a pair of decoupled second-order EPs, not a higher-order EP. So why point out the "second-order" aspect? If it's a tangential point related to future generalizations, it is best moved to the conclusion.*

Response 1.3: Thanks for your important advice. Indeed, to avoid potential confusion we have removed “second-order” in the revised manuscript as, “However, these efforts have always been limited to the exploitation of a single EP, restricting the response of the system to one single-polarization eigenstate defined by the handedness of the EP” in Line 18-20, Page 3.

Comment 1.4: *Line 83: "reflection/transmission zero singularities" -- the use of "reflection/transmission" is ambiguous about whether the zero occurs in the reflection*

AND transmission simultaneously, or reflection OR transmission. For clarity, this passage should be immediately followed up by a quick description of what these singularities mean, with minimal jargon.

Response 1.4: Thank you very much for pointing out this possible confusion. We have revised the description in the revised manuscript as, “A non-Hermitian metasurface, composed of subwavelength anisotropic structure arrays, can be designed to reach a degenerated eigenstate response at an EP, resulting in reflection singularities. This peculiar behavior is extremely interesting when it occurs on the polarization states, i.e., considering an EP of the Jones matrix, because the response of the interface totally vanishes for one of the two circular polarization (CP) states, leaving the other crossed CP channel unaffected with high reflection. In essence, these reflection singularities manifest as the zero values of the off-diagonal terms in the reflection Jones matrices. Also, we note that similar behavior can be observed for transmission zeros.” in Line 3-11, Page 4.

Comment 1.5: In the text accompanying the description of Fig. 1a, L_1 and L_3 are suddenly referred to, but the reader only finds out what these mean by looking at Fig. 1d. The subplots should be rearranged so that the order follows the logic in the text discussion.

Response 1.5: Thank you very much for your valuable suggestion. In order to make the plot arrangement more reasonable, we have made the following modifications to Fig. 1A and the corresponding descriptions:

Figure 1. Design principle of topological vectorial metasurfaces with EPs pair. (A) Perspective view of the structural unit of topological vectorial metasurface. The horizontal polarized light is perpendicular to the metasurface along the $-z$ direction. The inset figure on the upper right depicts the top view of the structure layer. The lengths of the three rods in the meta-structure are denoted by L_1 , L_2 and L_3 , respectively.

Comment 1.6: Line 125: "the degeneracy" is referred to without proper introduction/discussion of what is degenerate.

Response 1.6: Thanks for your important advice. In the main text, the first occurrence of degeneracy/degenerate happens in the sentence “leading to the potential coalescence of eigenstates as well as of eigenvalues at specific points, known as EPs. Such

degeneracy of eigenstates has aroused extensive interest in the asymmetric control of the eigenstates by researchers in various fields” in the **Introduction** (line 57). To eliminate the ambiguity in the description, we have made the following modifications in the manuscript:

“leading to the potential coalescence (i.e., **degeneracy**) of eigenstates as well as of eigenvalues at specific points, known as EPs. Such degeneracy of eigenstates has aroused extensive interest in the asymmetric control of the eigenstates by researchers in various fields” in Line 8, Page 3.

And:

“resulting in the flipping of the **degenerated eigenstate** from $E_x - iE_y$ to $E_x + iE_y$, i.e., from RCP to LCP (denoted as $|R\rangle$ and $|L\rangle$, respectively).” in Line 16, Page 6.

Comment 1.7: *Fig. 1i is only vaguely described in the text and the caption. This is not good, because it's directly related to the main selling point of "engineering arbitrary states", whereas the previous subplots, a-h, are more about controlling one polarization state (similar to earlier papers).*

Response 1.7: Thanks very much for this important comment that is essential for the reader to understand the novelty of the article. As you mentioned, **Fig. 1I** indeed holds a very important position in the paper as it represents our design idea. In the previous subplot, we designed and verified that a pair of meta-structures conforming to the rules of mirror symmetry can realize a pair of decoupled EPs. And it is not until **Fig. 1I** that they are truly combined together for the construction of the full polarization states. To better highlight our key idea and to comply with the referee’s suggestion, the manuscript has been modified as follows:

“**To realize asymmetric vectorial wavefront control for arbitrary polarization state reconstruction**, for LCP incident light we arrange the S meta-structures on a horizontal line and rotate them clockwise with the set initial orientation angle and an angle increment of φ_d for each adjacent meta-structure, i.e., to deflect the resulting RCP output light to the right side with a deflection angle of θ_t (**Fig. 1G**). Similarly, for RCP incident light, by rotating the S^m meta-structures counterclockwise with another initial orientation angle and setting the same angle increment φ_d , the output LCP can achieve the same deflection angle (**Fig. 1H**). **Importantly, taken into account that any polarization of the light beams can be described by a superposition of LCP and RCP (More details in Supplementary Note 3), a single metasurface composed of two types of crossed polarized meta-atoms has been used to project holographic images at arbitrary polarization. Here, we additionally preserve the asymmetric wavefront operation characteristics of EP designs by choosing two symmetrical EP meta-atoms, each respectively operating at one of the two orthogonal polarization states (Fig. 1I).** By encoding the holographic phase information on both LCP and RCP beams starting from the initial orientation angle and applying a modified Gerchberg–Saxton (GS) algorithm, arbitrary polarized meta-hologram can be achieved. It is worth noting that, due to the symmetric processing capability of the PB phase for circularly polarized light, the other CP beam generally appears in the $-\theta_t$ direction. However, the introduction of an EP pair totally suppresses redundant images in unnecessary directions, yielding

to asymmetric meta-hologram with single order. In brief, the design method uses the PB phase to realize the vectorial holographic images while avoiding unnecessary orders thanks to the degeneracy of the EP, providing a general strategy for asymmetric control of the vectorial wavefront.” in Line 10-30, Page 8 and Line 1-5, Page 9.

Comment 1.8: *The graphics in Fig. 1i itself are rather small and sparsely labelled. This is surprising since this is the main design idea for the experimental device! The key point to convey here is that there are two types of meta-atoms (targeting different EPs related by mirror symmetry) laid out on a single metasurface, with rotations to induce a PB phase in each channel.*

Response 1.8: Thanks for your important advice. In order to make the main design ideas in Fig. 1i more prominent, we have made corresponding modifications to Figs. 1G-I:

- 1) In Figs. 1G-I, the labels for the structures used in the figures are added, such as “Structure S ” in Fig. 1G.
- 2) In Figs. 1G-I, the labels have been added for the channels where the output beams are located, such as “ M_{RL} ” in Fig. 1G.
- 3) In Fig. 1G, the marker that LP-H is equivalent to RCP+LCP was added.
- 4) In Figs. 1G-I, the Poincaré sphere demonstration of the output polarization states has been added.

Figure 1. Design principle of topological vectorial metasurfaces with EPs pair. (G and H) Schematic of the asymmetric wavefront modulation by the combination of EPs and PB phase with rotating mirror-symmetric structures. The LP-H input light can be decomposed into RCP+LCP beams. The structures S in Fig. 1G work for LCP input light and the structures S^m in Fig. 1H work for RCP. The polarization states of the output beams are marked on the Poincaré sphere below, where the north and south poles

represent RCP and LCP, respectively. (I) Full-polarization-reconstruction through the superposition of output RCP and LCP beams at the EP pair with arbitrary phase and amplitude is realized by inputting the same LP beams. With the arrangement of two types of meta-atoms (S and S^m) on a single metasurface, arbitrary output polarization states can be realized.

In addition, as stated in **Response 1.7**, we have added a description of **Fig. 1I** in the main text:

“To realize asymmetric vectorial wavefront control for arbitrary polarization state reconstruction, for LCP incident light we arrange the S meta-structures on a horizontal line and rotate them clockwise with the set initial orientation angle and an angle increment of φ_d for each adjacent meta-structure, i.e., to deflect the resulting RCP output light to the right side with a deflection angle of θ_t (**Fig. 1G**). Similarly, for RCP incident light, by rotating the S^m meta-structures counterclockwise with another initial orientation angle and setting the same angle increment φ_d , the output LCP can achieve the same deflection angle (**Fig. 1H**). Importantly, taken into account that any polarization of the light beams can be described by a superposition of LCP and RCP (More details in **Supplementary Note 3**), a single metasurface composed of two types of crossed polarized meta-atoms has been used to project holographic images at arbitrary polarization. Here, we additionally preserve the asymmetric wavefront operation characteristics of EP designs by choosing two symmetrical EP meta-atoms, each respectively operating at one of the two orthogonal polarization states (**Fig. 1I**). By encoding the holographic phase information on both LCP and RCP beams starting from the initial orientation angle and applying a modified Gerchberg–Saxton (GS) algorithm, arbitrary polarized meta-hologram can be achieved. It is worth noting that, due to the symmetric processing capability of the PB phase for circularly polarized light, the other CP beam generally appears in the $-\theta_t$ direction. However, the introduction of an EP pair totally suppresses redundant images in unnecessary directions, yielding to asymmetric meta-hologram with single order. In brief, the design method uses the PB phase to realize the vectorial holographic images while avoiding unnecessary orders thanks to the degeneracy of the EP, providing a general strategy for asymmetric control of the vectorial wavefront.” in Line 10-30, Page 8 and Line 1-5, Page 9.

We believe we had successfully addressed all the comments by reviewer #1.

Reply report to reviewer #2

Comment 2.1: *The authors have addressed the technical issues I raised in my previous review. The experimental results clearly align with the theoretical calculations and the design.*

Response 2.1: We appreciate the reviewer for the positive assessment of our reply.

Comment 2.2: *In Comment 1.2, Reviewer #1 expressed doubt about the incremental technical advancement of the metasurface design using EPs. However, after reviewing previous works on polarization control using EPs, as well as the theories presented in the Supplementary Information, I am convinced that this manuscript offers a novel design principle and advanced technology that overcomes previous limitations.*

Response 2.2: We would like to thank the reviewer for the positive comments on the novelty of the design principle and technical advancement of our work.

Comment 2.3: *In Comment 1.3.2, Reviewer #1 expressed concern about whether this work would be of interest to a broader readership. However, I believe that Supplementary Note 2 provides sufficient information on the concept of EP pairs, which would be of interest to readers for those studying EP systems. In my opinion, the removal of twin images in metasurfaces is a significant challenge, and this work should be of broad interest to the metasurface community.*

Response 2.3: We appreciate the reviewer's statement on our work and the quality of our reply and associated modifications, and we hope this work can provide inspiration for future researches on EP systems and metasurfaces.

Comment 2.4: *Overall, I highly recommend this manuscript for publication in Nature Communications. However, I do have a few suggestions for the authors.*

Response 2.4: We are grateful to the reviewer's positive feedback and recommendation to publish in *Nature Communications*.

Comment 2.5: *It would be of interest to some readers to understand how the circular-polarization-based EP can be extended to achieve full-polarization reconstruction. I recommend including the statement from Response 2.3 in the Supplementary Information.*

Response 2.5: Thanks for your valuable suggestion. To better explain the reconstruction process of the full-polarization state, we have made appropriate modifications to the statement in **Response 2.3** and transferred it to the **Supplementary Note 3**.

“Supplementary Note 3: Reconstruction of the full-polarization state

The reconstruction of the full-polarization state in the far-field is achieved by superposing the scattered signal coming from a pair or more of phased controlled EPs. The phase adjustment is realized herein by choosing the initial orientation angles of structures S and S^m accordingly with the associated polarization dependent PB phase.

As mentioned in the manuscript, the role of structure S is able to convert LCP to RCP and its rotation assigns a PB phase equal to the twice of the rotation angle on RCP beam. Conversely, the structure S^m converts RCP to LCP and its rotation also assigns a PB phase equal to twice the rotation angle but with an opposite phase sign. The other channel of S and S^m (i.e., RCP incidence for S and LCP for S^m) is suppressed due to the reflection zero at EP and thus will not participate in the holographic imaging. Briefly, for S , the following relationship exists:

$$|L\rangle = e^{i2\varphi_R}|R\rangle \quad (\text{S40})$$

Also, for S^m , there exists:

$$|R\rangle = e^{-i2\varphi_L}|L\rangle \quad (\text{S41})$$

where φ_R and φ_L represent the initial orientation angles of S and S^m . To construct arbitrary polarization states covering the Poincaré sphere, the cases can be divided into the following three categories:

(1) When only one type of structure is used (as shown in **Figs. 3C** and **3D**), the polarization state of the output beam will be one of two circular polarization states with a specific phase related to the initial orientation angles of the structures (**Eqs. S40** and **S41**).

(2) When a pair of cross polarized EPs is superposed (as shown in **Fig. 3E**), the outgoing beam is a combination of RCP and LCP expressed by the following equation²:

$$|n\rangle = A_R e^{i2\varphi_R}|R\rangle + A_L e^{i-2\varphi_L}|L\rangle \quad (\text{S42})$$

where A_R and A_L are the amplitude of the RCP and LCP beams, respectively. The azimuth angle ψ and ellipticity angle χ of $|n\rangle$ can be calculated by $\psi = \varphi_R + \varphi_L$ and $\chi = \frac{1}{2} \arcsin \frac{A_R^2 - A_L^2}{A_R^2 + A_L^2}$. In the above case, since the number of structures S and S^m

and their scattering responses are equal, the values of A_R and A_L are equal. Therefore, the output beam will be coupled as linearly polarized light with azimuthal angle $\psi = \varphi_R + \varphi_L$ and ellipticity $\chi = 0$. It can be found that the initial orientation angle will affect the magnitude of the azimuth angle, thereby changing the polarization state.

(3) More degrees of polarization control can be obtained by controlling the number of rows of S and S^m . The output polarization would therefore be given by the superposition of the signal coming from each row. Take the simplest case as an example, that is, coupling two identically oriented S and one S^m (as shown in **Fig. 3F**) elements. In this case the value of A_R can be adjusted by the following formula:

$$A_R = \sqrt{(1 + \cos 2\Delta\delta_R)} \quad (\text{S43})$$

where $\Delta\delta_R$ represents the rotation angle difference of the two structures S . It indicates that the difference in the initial orientation angles of the two structures S has an impact on the ellipticity angle. In addition, as mentioned earlier, the azimuth angle is related to

the initial rotation angles of S and S^m . Therefore, by adjusting the initial orientation angles of structures S and S^m , changes in azimuth and ellipticity angles can be achieved simultaneously.

In fact, when two rows of S and two rows of S^m are combined as a unit, arbitrary adjustments regarding A_L and A_R can be achieved, resulting in an arbitrary distribution of ellipticity angles, leading to the reconstruction of full-polarization state.”

In addition, we have added the following description to the main text:

“By changing the number of rows of S and S^m to adjust the amplitude, and the rotation angle between the two rows to introduce the phase difference between LCP and RCP components (More details in **Supplementary Note 3**), we realized the holographic images with LCP (**Fig. 3C**), RCP (**Fig. 3D**), linear polarization (LP, **Fig. 3E**) and elliptical polarization (ELP, **Fig. 3F**.” in Line 5, Page 10.

Comment 2.6: *Many researchers studying vectorial holography may have the same question addressed in Comment 2.4. Therefore, I recommend including the statement from Response 2.4 and the simulated gray-scale vectorial hologram (Fig. R2-3) in the Supplementary Information. This would provide valuable additional information for readers interested in vectorial holography.*

Response 2.6: Thank you very much for your valuable suggestion. We have recognized that the explanation of vectorial holography in the article is indeed lacking. Therefore, we have integrated the content of **Response 2.4** and **2.9** and added it in the Supplementary Information:

“Supplementary Note 4: Specific polarization design of holographic images

In the analysis of **Supplementary Note 3**, we have confirmed that the structures shown in **Fig. 4A** can independently distribute arbitrary phase information to each pixel, which provides the basis for the control of far-field amplitude and polarization information.

To decouple intensity from far-field polarization information, we use a modified Gerchberg-Saxton (GS) algorithm² for vectorial fields. Assuming that the intensity of the far field is I^f , the azimuth angle is ψ^f , and the ellipticity angle is χ^f (superscript f represents far-field image plane), we can obtain the amplitude information of the LCP (a_L^f) and RCP (a_R^f) in the far field:

$$a_L^f = \sqrt{[I^f - I^f \sin(2\chi^f)]/2} \quad (\text{S44})$$

$$a_R^f = \sqrt{[I^f + I^f \sin(2\chi^f)]/2} \quad (\text{S45})$$

In addition, the phase difference between LCP and RCP (α^f) has the following relationship with the azimuth angle (ψ^f):

$$\alpha^f = 2\psi^f \quad (\text{S46})$$

Subsequently, a random phase φ_{rd} is given to the amplitude information described above to obtain the initial complex amplitude ($a_{L,R}^f e^{i\varphi_{rd}}$), and then the initial

metasurface information is obtained by the inverse Fourier transform:

$$B_L^m(1) = \mathcal{F}^{-1}(a_L^f e^{i\varphi_{rd}}) \quad (\text{S47})$$

$$B_R^m(1) = \mathcal{F}^{-1}(a_R^f e^{i\varphi_{rd}}) \quad (\text{S48})$$

Afterwards, the iterative Fourier transform process with the iteration number j from 1 to N is applied. It should be noted that the amplitudes of LCP and RCP in the metasurface plane are set as constants in each iteration.

If j is an odd number, the algorithm is described as:

$$\begin{cases} C_R^f(j) = \mathcal{F}(e^{i\angle[B_R^m(j)]}) \\ B_R^m(j+1) = \mathcal{F}^{-1}(a_R^f e^{i\angle[C_R^f(j)]}) \\ C_L^f(j) = \mathcal{F}(e^{i\angle[B_L^m(j)]}) \\ B_L^m(j+1) = \mathcal{F}^{-1}(a_L^f e^{i(\angle[C_R^f(j)] - \alpha^f)}) \end{cases} \quad (\text{S49})$$

And if j is an even number, the algorithm is:

$$\begin{cases} C_L^f(j) = \mathcal{F}(e^{i\angle[B_L^m(j)]}) \\ B_L^m(j+1) = \mathcal{F}^{-1}(a_L^f e^{i\angle[C_L^f(j)]}) \\ C_R^f(j) = \mathcal{F}(e^{i\angle[B_R^m(j)]}) \\ B_R^m(j+1) = \mathcal{F}^{-1}(a_R^f e^{i(\angle[C_L^f(j)] + \alpha^f)}) \end{cases} \quad (\text{S50})$$

where $\angle[D]$ represent the phase of D . When the iteration process ends, the final phase information of the metasurface is as follows:

$$\varphi_L^m = \angle[B_L^m(j)] \quad (\text{S51})$$

$$\varphi_R^m = \angle[B_R^m(j)] \quad (\text{S52})$$

Considering the relationship between the rotation angle of the meta-structure and the geometric phase, the rotation angle is determined as:

$$\Delta_L = -\frac{1}{2}\varphi_L^m \quad (\text{S53})$$

$$\Delta_R = \frac{1}{2}\varphi_R^m \quad (\text{S54})$$

Therefore, we can encode two different images as azimuth and ellipticity angles into a uniformly distributed intensity profile and achieve arbitrary combination of spatial variation amplitude and polarization information.

It is worth noting that since the number of rows of S and S^m in the metasurface is equal, there is no additional degree of freedom to control the intensity difference between LCP and RCP. We need to tune the orientation and ellipticity (translation and scaling) to make the total strength of LCP and RCP equal in the far field.

Furthermore, this Modified GS algorithm can be adapted to more complex situations, such as grayscale images. **Fig. S11** shows the asymmetric vectorial meta-hologram realized by grayscale images, in which the intensity, azimuth and ellipticity angles are all replaced by grayscale images. In **Fig. S11A**, the grayscale image of Van Gogh's self-portrait is used as intensity. **Fig. S11B** shows Van Gogh's "The Starry Night", and **Fig.**

S11C is a grayscale image of “Rest from Work”. The corresponding simulation results show that the meta-holograms are well reconstructed, confirming the potential of the Modified GS algorithm, which can achieve arbitrary combination of intensity, ellipticity and azimuth angles in the selected region.”

Supplementary Figure S11. Design of the spatially distributed polarization profile. (A to C) Designed vectorial hologram with the distribution of (A) intensity, (B) azimuth and (C) ellipse angle. (D to F) Simulated vectorial hologram with distribution of (A) intensity, (B) azimuth and (C) ellipticity angles of the polarization in the far-field.

And the following explanation has been added to the main text to enrich the content related to vectorial holography:

“By manipulating the amplitude (a_L , a_R) and phase difference (δ) of the two CPs over the far-field spatial distribution, arbitrary combinations of ellipticity and azimuth angles can be achieved in the selected region (**Supplementary Note 4**).” in Line 4, Page 11.

“Furthermore, grayscale images were used to demonstrate the enormous potential of the Modified GS algorithm. In **Fig. S11**, three grayscale images are selected as intensity, azimuth and ellipticity angles respectively, and the corresponding simulation results show that the meta-holograms are well reconstructed, indicating that the algorithm can achieve arbitrary combinations of intensity, azimuth and ellipticity angles in the selected area.” in Line 20-25, Page 11.

In summary, we have addressed all the comments of reviewer #2.

References

1. Hodaei, H. *et al.* Enhanced sensitivity at higher-order exceptional points. *Nature* **548**, 187-191 (2017).
2. Song, Q. *et al.* Broadband decoupling of intensity and polarization with vectorial Fourier metasurfaces. *Nature communications* **12**, 3631 (2021).